# Ultrasensitive nano-optomechanical force sensor operated at dilution temperatures

Francesco Fogliano[1], Benjamin Besga[1], Antoine Reigue[1], Laure Mercier de Lépinay[1], Philip Heringlake [1], Clement Gouriou [1], Eric Eyraud[1], Wolfgang Wernsdorfer [1], Benjamin Pigeau [1] & Olivier Arcizet[1✉]

Cooling down nanomechanical force probes is a generic strategy to enhance their sensitivities through the concomitant reduction of their thermal noise and mechanical damping rates. However, heat conduction becomes less efficient at low temperatures, which renders difficult to ensure and verify their proper thermalization. Here we implement optomechanical readout techniques operating in the photon counting regime to probe the dynamics of suspended silicon carbide nanowires in a dilution refrigerator. Readout of their vibrations is realized with sub-picowatt optical powers, in a situation where less than one photon is collected per oscillation period. We demonstrate their thermalization down to $32 \pm 2$ mK, reaching very large sensitivities for scanning probe force sensors, 40 zN Hz$^{-1/2}$, with a sensitivity to lateral force field gradients in the fN m$^{-1}$ range. This opens the road toward explorations of the mechanical and thermal conduction properties of nanoresonators at minimal excitation level, and to nanomechanical vectorial imaging of faint forces at dilution temperatures.

---

[1] Université Grenoble Alpes - CNRS - Grenoble INP, Institut Néel, Grenoble, France. ✉email: olivier.arcizet@neel.cnrs.fr

Operating force probes at low temperatures enables novel physical explorations in a quieter environment, with an increased force sensitivity granted by the reduction of their thermal noise which ultimately limits their performances. Since the invention of the atomic force microscope, the sensitivity of force sensors was greatly enhanced by using nanomechanical force probes. This enabled mechanical detection of ultraweak interactions such as the force exerted by single electronic spins[1] or molecules[2] and investigations of fundamental processes in condensed matter such as the interaction of a nanoresonator to a superconducting qubit[3]. At cryogenic temperatures, the reduced thermal bath fluctuations often comes along with a decrease of mechanical damping rates which further improves the probe sensitivity. All those benefits can be fully exploited only if vibrations of the nanomechanical force sensors can be readout without increasing their noise temperature.

This challenge becomes excessively difficult with ultrasensitive nanomechanical force sensors due to the desired decoupling from their mechanical support which is a general condition to achieve larger quality factors and to the extreme aspect ratios of the commonly employed ultrasensitive geometries in the form of nanotubes, graphene, or nanowires[4–7], but also trapped particles[8–11]. The overall reduction of the heat conductance at low temperatures, which even accelerates below 100 mK renders this objective even more difficult.

In this perspective, the choice of the readout probe is decisive and one should aims at minimizing its impact on the force sensor. Several approaches making use of electron based readout schemes were already implemented, using atomic quantum point contact[12], SET[13,14], squids[15–17], MW cavities[18,19], or propagating electron-beam[20,21]. However the spatial integration of the readout apparatus imposes serious geometrical constraints on the physics that can be investigated with such probes and in general does not allow preserving a scanning probe capacity.

In the optical domain, successful developments were realized by employing the toolbox of cavity optomechanics[22–26], making use of an enhanced readout efficiency enabled by the large cavity finesse. However, a natural drawback of this optical multi-path strategy is to increase the overall system absorption, which has rendered difficult the observation of the oscillator thermal noise thermalized at sub-100 mK temperatures. In those approaches, low phonon number occupancies were obtained by exploiting dynamical backaction cooling, which however does not necessarily help increasing the bare force sensitivity of the system since the active cooling process simultaneously reduces the mechanical susceptibility of the oscillator.

Several optical detection schemes making use of cavity-free optical interferometric readout were employed to probe ultrasensitive force sensors at low temperatures, in particular in the field of MRFM[27–30], but the thermal noise of the probe could not be measured below an effective noise temperature of ~110 mK. Despite the low light powers employed, the residual light absorption has thus for the moment prevented existing experiments from reaching the sub-100 mK temperature regime. This is however an extremely interesting objective since—beyond increasing the force sensitivity of the nano-optomechanical force probes- it permits investigating the intrinsic mechanical, thermal, and physical properties of the systems in their lowest excitation state and gives access to a new physical explorations.

Here we demonstrate the possibility to use an interferometric optomechanical readout scheme to probe the vibrations of an ultrasensitive nanomechanical force probe, a suspended silicon carbide nanowire, thermalized to the base temperature of a dilution fridge. We achieve very large force sensitivities in the 40 zN Hz$^{-1/2}$ range and a sensitivity to force field gradients of 20 fN m$^{-1}$ in 40 s, which compares favorably to previous works on

nanomechanical scanning force probes[16,17,27,29,30]. In our approach, the displacements of the nanowire vibrating extremity are readout using large numerical aperture optics enabling interferometric readout, designed for dilution temperatures. This lateral readout scheme provides a very good readout stability and preserves the bottom access to the nanowire vibrating extremity, thus maintaining a scanning probe capacity[31–39]. Part of the challenge resides in the thermalization of those ultra large aspect ratios nanomechanical force probes (aspect ratios up to 2000) which presents thermal resistances already above $10^{11}$ K W$^{-1}$ at 10 K, while our measurements suggests that it even increases by 3 orders of magnitude at sub-100 mK temperatures. This requires operating at ultralow photon fluxes, in a regime where all standard continuous detectors are totally blinded by their dark noise. Instead, we have developed a methodology based on avalanche single photon counters operated in the Geiger ("click") mode featuring ultralow dark count rates in the attowatt range, while our measurements are realized with collected photon fluxes of a few $10^3$ counts s$^{-1}$ (less than one photon detected per mechanical period). The demonstrated capacity to probe the thermal noise of our nanomechanical force probes allows the investigation of their thermal and mechanical properties in this unexplored temperature regime with minimal external perturbation. This approach opens the road toward vectorial imaging of ultraweak force fields at dilution temperatures, with applications in condensed matter physics, in MRFM explorations down to the single quantum spin level[1,33], or in cavity nano-optomechanics in the single photon regime[38].

## Results

**The experiment**. The nanomechanical force probes employed in this work are silicon carbide nanowires, see Fig. 1a, suspended at the extremity of a sharp tungsten tip. The samples employed, named A, B, and C are 330, 216 and 245 μm long respectively, with diameters of 300, 175, and conical from 240 to 120 nm (at the vibrating extremity), oscillating at 3.94, 5.84, and 11.6 kHz (see Supplementary Note 3). They are attached over tenth of microns on the side of a high purity sharp tungsten tip using a carbon glue and annealed at 700 °C, to increase the quality factors above ≈10,000 at room temperature. The tip is set in a highly conductive copper support and squeezed using an hydraulic press to ensure an optimal thermal contact and thermally connected to the mixing chamber of the dilution fridge using soft copper links. The vibrations of the nanowire are interferometrically readout by optical means as explained below. The experiment (see Fig. 1b–d) is mounted on the cold plate of a table-top wet dilution cryostat adapted to the requirements of ultrasensitive optomechanical experiments such as low injection pressure and reduced vibrations. A suspension apparatus featuring a 2 Hz cutoff frequency allows us to efficiently reduce the residual vibration noise originating from the boiling of the helium mixture in the still, down to a negligible level. The minimum mixing chamber temperature is of 20 ± 2 mK, while the mechanically decoupled experimental platform reaches a sample temperature of 27 ± 3 mK in operating conditions. It can be independently heated up to 500 mK without perturbing the dilution regime using a deported resistive heater.

The objectives are micro-positioned using piezo-electric 3D scanning-stepping stages while the sample is mounted on an XYZ piezo scanner offering 9 × 2 × 9 μm scanning window, at 4K and below, sufficient to explore the waist area.

**Optomechanical readout in the photon counting regime**. The interferometric objectives (see Fig. 1c, e) were specifically developed for low temperature purpose. The probe light (He–Ne laser, 632.8 nm) is focussed in a monomode fiber, carried into the

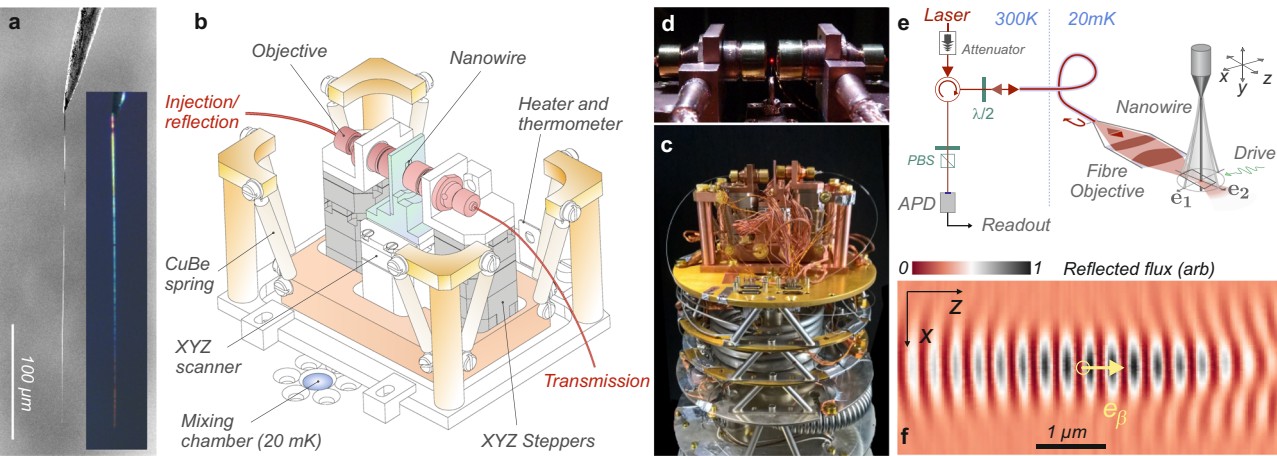

**Fig. 1 Experimental setup. a** SEM and optical images of the silicon carbide nanowires of large aspect ratios employed in this work. **b, c** Sketch and image of the experiment mounted on the cold plate of the dilution fridge: the light is fiber guided to the suspended cryogenic experimental platform, and focused on the nanomechanical oscillator under investigation. **d** Close up of the optical apparatus also showing a fraction of the focused light scattered by the nanowire. **e** Sketch of the cryogenic interferometric readout. Part of the light is reflected on the output face of the monomode fiber and interfere with the light back-scattered by the nanowire. **f** Moving the nanowire (2 μm above its vibrating extremity) in the waist area, in the *xz* horizontal transverse plane, generates a very contrasted reflectivity map, where the phase curvature of the focused laser beam can be clearly identified.

cryostat and to the cold plate where it is collimated and focused using two aspheric lenses, the front one featuring a 0.72 numerical aperture. This produces a 500 nm waist at a working distance of 3 mm with an overall transmission around 92%. About 4 % of the incident light is reflected on the fiber output face inside the fibered objective and interfere on the detector with the light reflected from the nanowire which is channeled back into the fiber. This defines an ultra-stable cryogenic interferometer (with a fringe stability of a few nm over days without active stabilization), featuring a relatively large spatial contrast for the nanowires employed (see Fig. 1f and Supplementary Note 4).

Moving the nanowire extremity in the optical waist allows us to spatially map the interference readout pattern $\Phi_R(\mathbf{r})$ measured in the reflection channel, as shown in Fig. 1f. The expected $\lambda/2$ axial periodicity is clearly visible and serves to calibrate the piezo displacements at different temperatures, which are typically reduced by a factor of 4 along the horizontal XZ axes between 300 K and 20 mK. Due to the nanowire sub-wavelength-sized-diameter, this also reveals the optical properties of the focused light beam, and in particular the wavefronts structure and possible deviations from the paraxial approximation. Due to internal optical (Mie) resonances, the optical contrast achieved (see Fig. 2a) varies with the nanowire diameter, the probe wavelength and its polarization, see Supplementary Note 4, but can be found close to unity which allows us to operate close to the dark fringe to increase the signal to background ratio (see below and Supplementary Note 6).

To avoid optical heating of the nanowires through residual absorption, we will see that it is necessary to operate at ultralow (sub-pW) optical powers, where the nanowire induced intensity fluctuations are likely to be hidden by the dark noise of standard continuous photodetectors. We circumvented this limitation by employing single photon counters, avalanche photodiodes operated in Geiger mode featuring a ~50 cts/s dark count rate, corresponding to ~15 attowatts in the visible and a quantum efficiency around 60%. The reflected signal recorded on the APD is thus constituted of a sequence of single photon pulses (see Fig. 2b), which can be either counted on photon counters to determine the mean photon flux (in cts/s) averaged over a measurement gate time, or analysed with time-stamping devices to acquire the full temporal statistics or with spectrum and

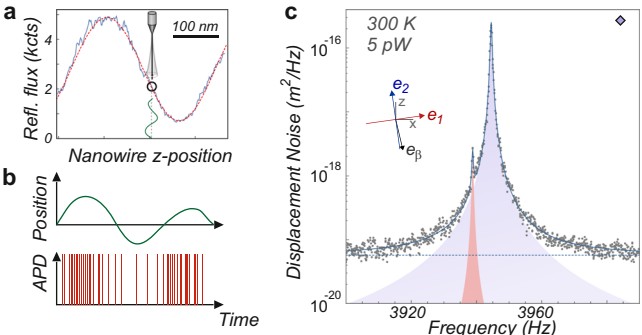

**Fig. 2 Nano-optomechanical readout in the photon counting regime. a** When the nanowire is placed at a position presenting a large spatial variations of the reflected flux, its position fluctuations are converted into a large flux modulation (**b**). The SNR can be improved by working in the lower part of the interference slope, to reduce the shot-noise level. **c** Typical thermal noise spectrum (nanowire A) obtained at room temperature with ≈5 pW of injected light, used to determine the effective mass (16 pg) and the eigenmodes orientations (inset) of both nanowire transverse fundamental eigenmodes.

network analysers to measure the nanowire thermal noise spectra or its response to an external driving force.

As sketched in Fig. 2a, b, when the nanowire extremity is positioned on the slope of the interference pattern $\Phi_R(\mathbf{r})$, its transverse vibrations $\delta\mathbf{r}(t)$ around the rest position $\mathbf{r_0}$ are dynamically encoded as temporal variations of the reflected photon flux $\Phi_R(\mathbf{r_0} + \delta\mathbf{r}(t))$ according to: $\delta\Phi_R(t) = \nabla\Phi_R|_{\mathbf{r_0}} \cdot \delta\mathbf{r}(t)$. This provides a projective measurement channel of the nanowire vibrations $\delta r_\beta(t) = \mathbf{e}_\beta \cdot \delta\mathbf{r}(t)$, projected on a normalized measurement vector defined by $\mathbf{e}_\beta \equiv \nabla\Phi_R|_{\mathbf{r_0}}/|\nabla\Phi_R|_{\mathbf{r_0}}|$. The latter is measured using a routine computing the local tangent plane to the surface $\Phi_R(\mathbf{r})$ as illustrated in Fig. 1f (see Supplementary Note 4). The calibration of the spectral measurements performed with avalanche detectors is realized in comparison to the calibrated mechanical noise spectra obtained using standard photodiodes and well established calibration methods[32,35], following a methodology exposed in the Supplementary Note 5. We determine in particular

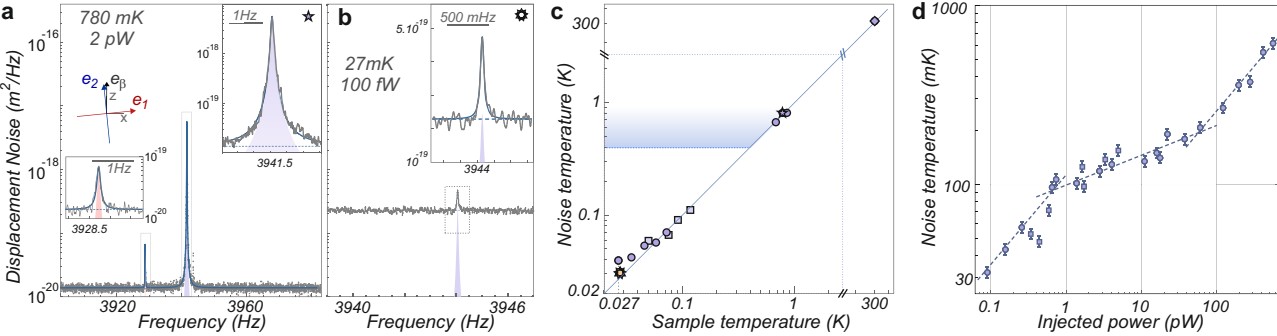

**Fig. 3 Noise thermometry. a** Calibrated thermal noise spectrum obtained on nanowire A for an experiment temperature of 780 mK. The eigenmode orientations and effective mass are unchanged, while the quality factor is increased to ≈80,000. **b** Similar measurement obtained for an injected power of 100 fW at 27 mK. **c** Vibration noise temperature, $T_{eff}$ as a function of the sample temperature. Different measurements are shown, with suspensions (star/squares) and without while stopping (circles) the mixture injection, using optical powers in the 100–300 fW range. The lowest noise temperatures obtained are around 32 ± 2 mK, for a sample temperature of 27 mK and a base temperature of 20 mK. The shaded area indicates the noise temperatures measured in absence of suspension. **d** Dependence of the nanowire noise temperature $T_{eff}$ measured for increasing injected optical powers while maintaining the cryostat at its minimal temperature. The dashed lines are indicative power laws of 1/2, 1/5, and 1/2, to illustrate the different regimes of sublinear heating rates observed.

the conversion coefficient $\eta = 3.98 \times 10^8$ V/(cts/s), which depends on the pulse shape and electrical impedances. It enables the conversion of the noise spectral density $S_V[\Omega]$ measured on a spectrum analyzer into a projected displacement noise spectral density: $S_{\delta r_\beta}[\Omega] = S_V[\Omega]/(\eta^2|\nabla\Phi_R|_{r_0}|^2)$, that contains the nanowire thermal noise $S_{\delta r_\beta}^{th}[\Omega]$ which will serve to assess the vibrational noise temperature and a shot-noise limited background $S_{\delta r_\beta}^{shot} = \Phi_{r_0}/\left|\nabla\Phi_R|_{r_0}\right|^2$. This expression allows us to determine the position in the interference pattern which maximizes the signal to background ratio. It is in general not found at the position of maximal slope, but closer to the dark fringe (see Fig. 2). In the case of a perfectly dark fringe, it is possible to gain up to 3 dB on the signal to shot-noise ratio, which can be appreciable at low temperatures.

**Noise thermometry**. The following measurements were realized on nanowire A (330 μm long, 300 nm diameter), featuring an effective mass of 16 pg and a quality factor around $10^4$ at 300 K. A typical thermal noise spectrum obtained at room temperature is shown in Fig. 2c. Despite the low optical power employed, 5 pW of injected light (measured at the fiber input), the two fundamental transverse eigenmodes emerge with a large dynamics (37 dB) above a flat shot-noise limited background $S_{\delta r_\beta}^{shot}$. The mechanical thermal noise spectra can be fitted with the expression:

$$S_{\delta r_\beta}[\Omega] = \sum_{m=1,2} \frac{2\Gamma_m k_B T_{eff}^m}{M_{eff}} \frac{(\mathbf{e}_m \cdot \mathbf{e}_\beta)^2}{(\Omega_m^2 - \Omega^2)^2 + \Gamma_m^2 \Omega^2} + S_{\delta r_\beta}^{shot} \quad (1)$$

to extract the nanowire noise temperature ($T_{eff}^m$), where the other fitting parameters are the shot-noise level, mechanical frequencies $\Omega_{1,2}/2\pi$, damping rates $\Gamma_{1,2}$ and eigenmode orientations $e_{1,2}$ (shown in the inset of Fig. 2c). The nanowire effective mass $M_{eff}$ was determined using the above expression after having verified the proper thermalization of the nanowire and the absence of optical heating or optical backaction[32,36] on the system, which are both largely negligible at room temperature as long as the injected optical power remains lower than a few μW. The typical variability on the measured $T_{eff}/M_{eff}$ ratio is around 5 %, and mainly depends on the imprecision of the measurement vector, which thus requires a careful estimation. At ultralow photon fluxes the larger relative fluctuations in the measured photon counts lead to

longer averaging times (tens of seconds) to reach a sufficient precision on the measurement vector.

During the cool down phase, we need to compensate for thermal drifts (tens of μm) by tracking the nanowire position and maintaining it in the waist area using the stepper motors supporting the objectives. The optical interference pattern remains unchanged at cryogenic temperatures (besides the change in the piezo expansion coefficient). Figure 3a shows the thermal noise spectra obtained for an intermediate temperature of 780 mK. The same fitting procedure permits to verify that the eigenmodes' orientations remain unchanged at cryogenic temperatures, while the quality factors are increased by a factor of 10. The effective mass is also unchanged within our experimental precision meaning that the eigenmodes longitudinal spatial profile is not altered, as expected for a singly clamped nanowire. Figure 3b shows the thermal noise spectrum obtained at a sample temperature of 27 mK for an injected power of 100 fW. In that situation, the collected optical flux amounts to a few kcts/s, comparable to the mechanical frequency, meaning that this measurement is realized with only ~1 photon collected per mechanical period. Despite this ultralow flux, it is still possible to preserve a signal to shot-noise ratio slightly above unity. Those measurements are realized with a resolution bandwidth of 30 mHz and averaged for 30–60 min. Due to the modest signal to background ratio obtained, it can be delicate to properly estimate the nanowire quality factor (but it can be better evaluated using driven measurements, see below). To evaluate the corresponding imprecision in the noise temperature evaluation, we fit the data by imposing a ±5% variation of the damping rate around the best fit value, giving a variability of ±5% on the fitted temperature (see Supplementary Note 11). Those values are in agreement with the statistical error we obtained on iterative measurements, while the overall absolute imprecision is limited by the one obtained on the measurement slope to a ±2.5% level. As a result, we estimate the effective temperature of the nanowire vibration noise at a level of 32.1 ± 1.8 mK, for a sample temperature of 27 ± 3mK, measured with a RuO₂ thermometer (and a 22 mK mixing chamber temperature).

Achieving a noise temperature comparable to the cryostat base temperature requires minimizing the optical absorption (as discussed below) and other sources of heat impinging on the sample, but also taking a special care to minimize electrical and mechanical parasitic noises, which can drive the nanowire to an artificially large noise temperature. The electrical signals carried

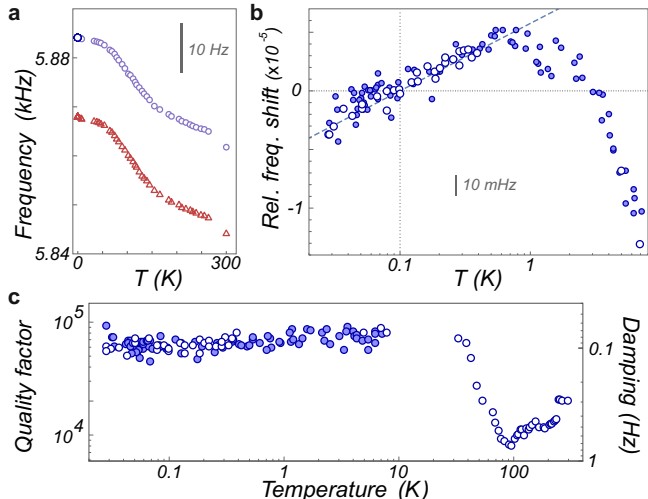

**Fig. 4 Mechanical properties. a** Temperature dependence of the eigenfrequencies of the 5.8 kHz nanowire (B). **b, c** Evolution of the relative frequency shifts and mechanical damping rates (mode 2) below 10 K. The open circles are obtained at minimal optical powers (sub pW), while varying the cryostat temperature. The full disks represent the frequency shifts and damping rates measured for increasing optical powers (from 100 fW to 10 nW, cryostat set at 20 mK), as a function of the measured effective noise temperature (after subtraction of a 30 mK of artificial noise contribution). The dashed line in (**b**) corresponds to $\Delta f_2/f_2 = 2.5 \times 10^{-6} \ln(T/100\,\text{mK})$.

down to the piezo stages have to be strongly filtered (1.6 Hz bandwidth) and if possible grounded to avoid unwanted electrically driven vibrations of the nanowire support. To mitigate vibrational noise, the cryostat is mechanically decoupled from the building and the gas handling system (see Supplementary Note 1), but we initially suffered from the acoustic noise generated by the circulation of the helium mixture (400 mbar injection pressure). This caused a vibration noise of the nanowire support of a few pm Hz$^{-1/2}$ at 4 kHz, which is relatively weak, but sufficient to resonantly drive the ultrasensitive nanowire, to an effective noise temperature in the 0.2–4 K range depending on the circulation conditions (dashed line in Fig. 3c). Running a measurement sequence in which the circulation was stopped for a short duration (1 h), sufficient to acquire meaningful spectra (circles in Fig. 3c), evidenced the impact of the circulation noise and noise temperatures down to 40 mK could be measured. We thus developed the suspension mechanism (see Fig. 1) with a cutoff frequency around 2 Hz, which allowed mitigating the circulation noise to a negligible level. Noise measurements conducted at different cryostat temperatures are shown in Fig. 3c and demonstrate the proper thermalization of the nanowire to the platform temperature.

**Mechanical properties.** The temperature dependence of the nanowire mechanical properties are shown in Fig. 4. Although the SiC nanowires are dominantly crystalline (3C phase), the temperature dependence of their mechanical properties presents large similarities with the universal properties observed on resonators made of amorphous materials[40–42]. The relative mechanical frequency shift increases logarithmically at low temperatures and presents an inversion above 700 mK. Also, the quality factor presents a minimum around 80 K and reaches a plateau below 10 K (the absence of quality factor improvement below 1 K is likely due to limiting clamping losses). However if the trends look similar, the observed magnitude is smaller. In particular the quality factors observed on the plateau are around 80,000, about

30 times larger than the universal value for amorphous materials of ∼2500. Similarly the relative frequency shifts follows $\Delta\Omega_\text{m}/\Omega_\text{m} \approx C\ln(T/T_0)$, with a constant $C \approx 2.5 \times 10^{-6}$, which is 2 orders of magnitude smaller than the universal value[40–42], which is directly connected to the density of structural defects involved. A simple interpretation could be that only a part of the nanowire behaves as an amorphous material, which is in agreement with the few nanometers oxide crust covering their surface, as observed in TEM imaging[43]. We observe that the temperature dependence of the mechanical frequency can be practically used as a very good indicator of the nanowire internal temperature and can serve to localize regions of minimum absorption on the nanowire surface. We note that if the nanowire is artificially driven by a parasitic force noise, this will cause an increased mechanical power dissipated by mechanical friction in the nanowire. However it remains largely negligible ($\Gamma_\text{m}k_B T_\text{eff} \approx yW$ for 1 K noise temperature) compared to the injected optical powers. This explains why, prior to the shielding of mechanical and electric noises, a nanowire could be found with a frequency indicating a good thermalization even at the lowest cryogenic temperatures despite presenting a large artificial excess of vibration noise.

**Optical heating.** The above measurements were conducted at minimal optical powers to avoid optical heating through residual light absorption. Depositing heat at the extremity of the nanowire while measuring the induced noise temperature increase can be used[44] to investigate the thermal properties of the nanowire at dilution temperatures. Figure 3d shows the nanowire noise temperature measured for increasing optical powers, spanning over 4 orders of magnitude up to 0.6 nW, causing a non-linear temperature increase up to 650 mK with 3 different regimes. A rapid increase, scaling as $P^{1/2}$, is observed below 100 mK, it flattens to approx. $P^{1/5}$ between 100 and 200 mK and accelerates again as $P^{1/2}$ above 200 mK. All those regimes present a sublinear dependency with the injected optical power, a consequence of the increase of the nanowire thermal conductance with temperature. A conductance scaling as $T^\mu$ will produce a $P^{1/(1+\mu)}$ optical heating trend (see Supplementary Note 7). Similar behaviors were observed on several nanowires (see Supplementary Fig. 8) and we note that the heating efficiency varies with the laser position on the nanowire which can be attributed to local variations in the absorption coefficient $A_\text{abs}$ (possibly due to contaminants). We also systematically observed that positioning the readout laser closer to the nanowire extremity (within 2 μm) also largely increases (×5 typically) the laser heating efficiency. This is attributed to a wave guiding mechanism (see Supplementary Fig. 7) which increases the amount of light guided inside the nanowire that can be subsequently absorbed. In view of the relatively long nanowires employed, we can position the readout laser at the least absorbing location in an extended area close to their vibrating extremity without suffering from a significant increase of the effective mass (see Supplementary Fig. 2). For large optical powers, around 100 nW, the vibration noise temperature can be raised up to 10 K, see Supplementary Fig. 8.

Those measurements can be exploited to deduce the variations of the nanowire conductance $K(T)$ along the optically induced temperature increase. First, as shown in Fig. 4 one can notice that heating the nanowire with the cryostat at low optical power or with the laser generates the same evolution of mechanical properties with the cryostat or noise temperatures respectively[44]. In the limit where the phonon mean free path remains small compared to the nanowire length, which occurs in principle only above 150 mK, see Supplementary Note 8, a temperature can be defined quasi-continuously along the nanowire (otherwise the

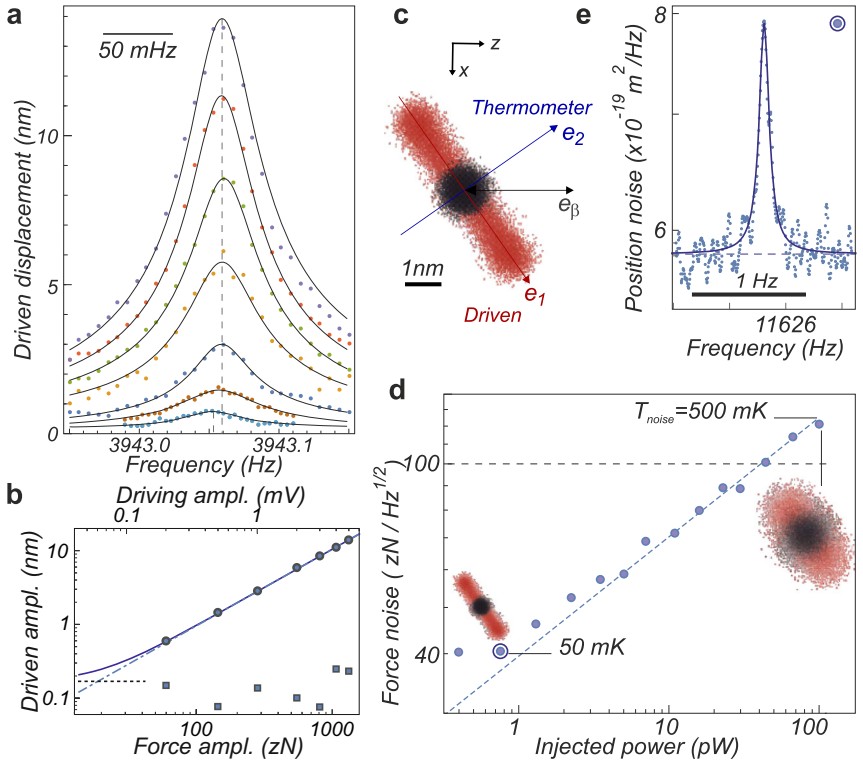

**Fig. 5 Force sensing. a** Response measurements obtained for increasing piezo actuation voltages (50 mHz resolution bandwidth). The fitted resonant amplitudes and background values (not rms) are reported in (**b**) as circles and squares respectively. The linearity of the system is well verified, with driving force amplitudes spanning from 59 zN to 2.9 aN ($\Gamma_m/2\pi = 40$ mHz here). The solid line represents $\sqrt{(\chi_2 \delta F)^2 + \mathcal{N}}$ with $\chi_2 = 1.0 \times 10^{10}$ m N$^{-1}$ where the displacement noise power $\mathcal{N}$ integrated over the resolution bandwidth is indicated as a dashed line. **c** Sketch of the measurement configuration realized on a 11 kHz nanowire: mode 1 is resonantly driven to an amplitude of 1.5 nm (1.8 aN drive amplitude), while the noise temperature is simultaneously measured on mode 2. The red and black trajectories illustrate the driven and non driven situations. **d** Dependence of the force noise of eigenmode 2 for increasing optical readout powers, while resonantly driving mode 1. The dashed line is a $P_{opt}^{1/2}$ guide line. The noise temperatures are indicated for 2 points in that plot, see Supplementary Note 9. **e** Corresponding thermal noise spectrum obtained using 750 fW, featuring a 50 mK noise temperature corresponding to a 40 zN Hz$^{-1/2}$ force sensitivity.

phonons may travel along without interacting). Then, when the conductance strongly increases with temperature, the temperature gradients in the nanowire become largely localized close to the supporting tip, which is assumed to be thermalized to the cryostat. The internal temperature profile is then expected to be almost homogeneous within the vibrating part of the nanowire and in first approximation, can be assimilated to the measured thermal noise temperature. This remark explains the similar behavior observed between both measurements in Fig. 4.

One can then adjust the optical heating curves to extract the thermal conductance, exploiting the relation $K(T)/A_{abs} = (dT_{eff}/dP_0)^{-1}$ or an equivalent integral formulation given in the Supplementary Note 7. We assume here that the optical absorption coefficient $A_{abs}$ does not vary with temperature and determine its value by setting the nanowire conductance at 10 K to the one given by the Casimir model, which accounts for the fact that the phonon mean free path is limited by incoherent reflections on the nanowire diameter, see Supplementary Note 8. Having $K(10\,K) = 3.5$ pW K$^{-1}$, we obtain $A_{abs} = 70$ ppm. Since the probe laser wavelength is significantly larger than the optical bandgap of 3C silicon carbide (515 nm), it is likely that the optical absorption is dominated by local contaminants.

The estimated conductance/absorption ratio, see Supplementary Fig. 8, is reduced by 4 orders of magnitude at 30 mK, down to 0.4 fW K$^{-1}$. This extremely weak magnitude explains the rapid laser heating induced by the probe laser and the difficulty to

observe a nanowire thermalized to the cryostat temperature, which required operating with sub-pW injected optical power. According to the mean free path analysis exposed in the Supplementary Note 8, below 150 mK the heat propagation becomes ballistic over the entire nanowire length, a regime which will be the topic of future investigations.

**Force sensing.** The mechanical response of the nanowire, measured along a measurement direction $\mathbf{e}_\beta$ to an external coherent driving force $\delta\mathbf{F}(t) = \delta\mathbf{F}\cos\Omega t$ exerted at the nanowire vibrating extremity[45] can be written, in the Fourier domain:

$$\delta r_\beta[\Omega] = \sum_{m=1,2} \frac{(\mathbf{e}_m \cdot \mathbf{e}_\beta)(\mathbf{e}_m \cdot \delta\mathbf{F} + \delta F_m^{th})}{M_{eff}(\Omega_m^2 - \Omega^2 - i\Omega\Gamma_m)} + \delta r_\beta^{shot}, \quad (2)$$

where the projected driving force adds up to the Langevin force noise $\delta F_m^{th}$, m = {1, 2}, whose noise spectral density $S_F^m = 2M_{eff}\Gamma_m k_B T_{eff}^m$ sets the force sensitivity of the nanowire in the framework of the linear response theory. The mechanical response of the nanomechanical force probes was tested under external drive by means of time-modulated piezo, electrostatic or optical actuations. We measured that they linearly respond to the external drives, from 59 zN up to relatively large force amplitudes, of a few aN, see Fig. 5a, b, which produce displacement amplitudes in excess of 10 nm. The absence of significant mechanical frequency shifts for increasing actuation strength suggests that the

forces employed to coherently drive the nanowire do not increase its bulk temperature.

To further investigate this question, we studied the nanowire noise temperature in presence of external actuation by resonantly driving one of its fundamental eigenmode, while realizing noise thermometry along the second perpendicular eigenmode, as sketched in Fig. 5c. We found that it was possible to observe the 11-kHz nanowire (sample C, 1.6 pg effective mass) thermalized down to 47 mK without suffering from additional unwanted heating due to the actuation mechanism, despite resonantly driving the first mode to a 3 nm amplitude. Those measurements also allowed to report the lowest thermal force noise measured, see Fig. 5d, e, at the level of $\sqrt{2M_{\mathrm{eff}}\Gamma_m k_B T_{\mathrm{eff}}} = 40$ zN Hz$^{-1/2}$ when thermalized at $T_{\mathrm{eff}} = 47$ mK, where $T_{\mathrm{eff}}$ and $\Gamma_m$ are deduced from fits of the thermal noise spectra using Eq. (1). This value remains rather large compared to the one reported in single ions[46,47], nanospheres[11] or nanotube[6] experiments, however the achieved force sensitivity compares favorably to previous approaches in scanning force sensors, even if the quality factors (90,000) were still limited by amorphous-like damping and clamping losses. To put this number in perspective, 40 zN corresponds to the Coulomb force between two electrons 70 μm apart.

The above observations also present an important practical interest for ultrasensitive vectorial force field sensing[34–36] since the nanowires can be resonantly driven up to large oscillation amplitudes, providing a large SNR on the readout channels, without seeing their low noise properties being degraded. The frequency stability of the nanowires was then investigated under similar driving conditions (see Supplementary Fig. 10). We measured relative r.m.s. frequency deviations around $4 \times 10^{-8}$ over hours and a relative Allan deviation of $5 \times 10^{-11}$ for gate times of $\tau = 2\pi/\Gamma_m$. This intrinsic stability sets a limit on the smallest lateral force gradients which can be detected through the frequency shifts and eigenmode rotations they produce[35], to a level of ~1 fN m$^{-1}$ (see Supplementary Note 10). This faint magnitude is comparable to the maximum force gradient (2 fN m$^{-1}$) produced by the static radiation pressure force exerted by the 100 fW readout laser (0.6 zN varying over 300 nm laterally)[32], underlying the importance to operate with ultralow readout powers in order to mitigate the probe backaction and preserve the ultimate performances of the ultrasensitive nano-optomechanical force probe.

As a conclusion, we have shown the possibility to readout the vibrations of ultrasensitive nanomechanical force probes down to the base temperature of a dilution fridge, providing large force sensitivity for a scanning probe force sensor. We used readout techniques operating in the photon counting regime, were less than a pW is injected in the system and less than one photon is detected per mechanical period in order to prevent heating via residual light absorption and reduce the probe backaction.

Their very large force sensitivity combined with a certain versatility opens the road towards vectorial imaging of weak forces in condensed matter systems, in proximity force measurements or experiments at dilution temperature aiming at exploring the single photon regime of cavity nano-optomechanics[38] or the spin qubit-nanoresonator interaction[33,48,49].

Further investigations will also aim at exploring the onset at low temperatures of the ballistic regime of heat transport along the nanowires[50]. In particular, measurements based on pump–probe responses with excitation and detection lasers located at different positions along the nanowire[51–54] will help exploring those transitions.

## Data availability
The data that support the findings of this study are available from the corresponding author upon reasonable request.

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

## Acknowledgements

We warmly thank N. Roch, F. Balestro, C. Winkelmann, J. Renard, and L. Marty for in house cryogenic developments and experimental assistance, the PNEC group at ILM, O. Bourgeois and E. Collin for fruitful discussions as well as J.P. Poizat, A. Auffeves, G. Bachelier, J. Jarreau, C. Hoarau, C. Felix, and D. Lepoittevin. F.F. acknowledges funding from the LANEF (ANR-10-LABX-51-01). P.H. acknowledges funding from the European Union H2020 program (Marie Sklodowska-Curie grant 754303). and QUENG. This project is supported by the French National Research Agency (JCJC-2016 CE09-2016-QCForce, SinPhoCOM, LANEF framework (ANR-10-LABX-51-01, project CryOptics and Investissements d'avenir program (ANR-15-IDEX-02, project CARTOF), and by the European Research Council under the EU's Horizon 2020 research and innovation program (AttoZepto 820033 project).

## Author contributions

All authors contributed to every aspect of the work. W.W. and E.E. designed and realized the cryostat. F.F., B.B., A.R., P.H., L.M., C.G., B.P., and O.A. worked on the cryogenic nano-optomechanical experiment.

## Competing interests

The authors declare no competing interests.
