## [Peer Review File · Nature Communications]

Reviewer #1 (Remarks to the Author):

The manuscript describes experimental results on vibrations of suspended SiC NWs excited by low optical power in a dilution refrigerator. The displacement of the NWs extremity was used as readout using numerical aperture optics. The results showed a force resolution of $40 \text{ zN/Hz}^{1/2}$ and a force sensitivity in the range of fN/m . I have no doubt that observation of the oscillator thermal noise at below 100 mK temperature could be beneficial to investigating the intrinsic mechanical and physical properties of the material and structure. Also, I have no criticisms on the experimental data.

It is not surprising that a high force sensitivity is observed at a low temperature of sub 100 mK. Even though this is a claim of record sensitivity for mechanical scanning force probes, I could not see where the results of this work stand in the field. Perhaps, systematic comparison of the results to the literature will help if advances have been made by this work.

Besides experimental capability of testing at sub 100 mK, I am not really happy about the scientific aspect of this work. Understanding the phenomenon behind is important and interesting, but being treated unequally. In addition to the size of the nanowire, material is the key factor determining the response of the optomechanical force sensor.

The mechanical properties of SiNWs were observed as large similarities with those in amorphous materials (figure 4). Did the author confirm if SiC NWs was amorphous? In some discussion, the authors mentioned cubic silicon carbide (3C). Are there any differences in the behaviour of the mechanical system if there are a change in material structures?

From engineering point of view, SiC NWs are selected but crucial material characteristics that are beneficial for optomechanical force sensors could be discussed. Material structures ranging from amorphous, nanocrystalline, polycrystalline and single crystalline with different doping levels could affect the mechanical and electrical properties, as well as the interaction between optical input and NWs structures, especially at low temperatures.

As thermalized down to sub 100 mK, noise temperature from unwanted heating could be significant. External drives could increase the stiffness and vibration frequency, while thermal heating has an adverse effect. Perhaps, location of excitation on the SiC NWs could be considered if there are considerable mechanical frequency shifts as heating has more significant impact toward the fixed end. This needs clarification.

Some minor points are as follows. Please proofread the manuscript, some typos are detected, e.g. 700 C; make clear references to the used equations; to make sure experiments are well controlled, discussions should be given on selection of vibrating frequency ranges and quality factor; information about material synthesis would be informative.

Overall, the authors present nice experiments and results. I advise publication of this work if the above points have been addressed. It is interesting to prove and clarify the science behind achieving ultra-high sensitivity of optomechanical force sensors. I would expect better guidance for researchers in design system for ultrasensitive force measurements.

Reviewer #2 (Remarks to the Author):

This work describes the characterization of SiC nanowire probes at millikelvin temperatures for eventual use as force and force gradient sensors in scanning probe applications. Because the optical readouts used for these applications usually generate more heat than a dilution refrigerator can dissipate, the authors propose a single photon interferometric readout scheme that allows them to drop their photon fluxes below about $3E6/s$ (~ 1 pW). They use this capability to measure the mechanical properties of the nanowire cantilevers as a function of temperature, establishing a consistent value for cantilever effective mass regardless of whether the nanowire was heated by changing the temperature of the cryostat or by changing the photon flux. The latter technique also allows them to make some general statements about the thermal conductivity of the SiC nanowire. The optical properties of the nanowire are potentially obscured by the surface contaminant layer. They then make some claims about the sensor's force and force gradient resolution in conclusion. In general, this is very good work. The experiment was challenging, and the authors appear to have conducted it with attention to detail in both the lab and in the theoretical approach. There is clear evidence of interference fringes in Figure 1, and although a similar approach has been used previously to extract the interferometer sensitivity, it has not been done with single photon detection. The characterization of the material properties of the nanowire itself are likely to have less impact, as the authors acknowledge the presence of an oxide layer and surface contaminants make the results not straightforward to interpret. In addition, several points about the claimed force resolution must be clarified before the work is ready for publication.

Specific Comments:

Page 1, Right Column, Paragraph 3: The use of the phrase "cavity-free" is not entirely justified. The authors have created a low-finesse optical cavity with the fiber face as one end and the nanowire as the other end. A cavity of some kind is necessary for this type of homodyne interferometer. It would be more accurate to say this system is free of a resonant cavity.

P2, L Column, Pgph1: The phrase "length over diameter" is more succinctly expressed as "aspect ratio."

Figure 1: A few of comments about this.

1. Please clearly label the coordinate system inset into Fig. 1e. It is difficult to parse Fig 1f without an ambiguous visual cue as to which direction x and z refer to. In general, z often aligns to gravity, but that would not make sense in light of the diagram in Fig. 1e and the interferogram in Fig. 1f.
2. I notice a PBS and two APDs in the experimental diagram of Fig 1e. Was this a Hanbury-Brown-Twiss interferometer? A brief explanation of the dual APD setup is warranted.
3. Would like some clarification about which part of the nanowire is generating the contrast. It looks like the center of the fringes in Fig.1f tail off at higher and lower values of x. The text hints that this is at the far distal end of the nanowire, but this should be clarified.

P3, L, 2: Relatively high NA objectives are used in this work. This severely restricts the working volume accessible to the experimenter, as any obstruction of the optical path will cause diffraction of the sensing laser used in the interferometer. Given that the nanowires are only a few hundred micrometers long, it will be extremely difficult to get a surface into close proximity. This also relates back to my question above about where on the nanowire the interferogram of Fig 1f is measured. Many of the fields that are of most interest to scanning probe microscopists decay rapidly with distance. If the authors want to use scanning probe microscopy as the primary justification for the relevance of their work, they should provide a realistic estimate of the spatial resolution they can

expect to achieve with the sensor during operation in a scanning probe mode.

Figure 3: Figure 3a, are the two peaks in this subfigure intended to represent two different vibration modes? If so, the color coding should match the inset and be consistent with figure 3 c and d.

P5, L, 1: Please clearly explain the distinction between nanowire temperature and sample temperature.

Figure 4, in Fig. 4a, presumably the two colors are meant to represent the two transverse vibrational modes. Optional suggestion: make different shapes for clarity in case the information is displayed in a format without color.

P6, L, 2: There is a lot of information crammed into this paragraph. From briefly reading reference 39, I think that the statement "a temperature can be defined quasi-continuously along the nanowire" needs to be clarified. Is this intended to state that the temperature gradient is nonzero and continuous along the length of the cantilever, or that the cantilever is at a constant uniform temperature along its length? Explain more how this contrasts with the case above 150 mK? Although this may be explained more fully in the supplement, a better explanation in the body of the paper is necessary.

P6, R,1: If the optical absorption is dominated by contaminants, what is to say that the thermal conductivity is not also? The arguments addressed in the previous comment (P6, L, 2) fall apart if there is something with much higher thermal conductivity coating the outside of the cantilever.

Fig 5. In figure 5b, the minimum driven amplitude (RMS, as I assume this is measured with a lockin amplifier, given the units) is approximately 1 nm. From the mass and first resonant frequency, I deduce that the stiffness of the relevant vibrational mode of the nanowire is $2.5E-6$. At resonance, the response of the nanowire cantilever is enhanced approximately by a factor of Q relative to the quasistatic case. I deduce Q to be approximately $1e5$ from the data in figure 5b, which leads me to conclude the minimum force measured by nanowire A is approximately 200 zN based on the data available. The effective mass of nanowire C is not reported. The information justification in the text for the putative $40 \text{ zN/Hz}^{1/2}$ noise floor is not clear. According to the supplement, the measurement stage was not suspended, and there appears to have been some background subtraction that is not adequately described in the text. The appropriate method for judging claims of minimum force noise in published literature (especially in high impact journals) is that the authors clearly provide an easily understandable justification for their claims without reference to the supplementary materials (which, in this case, are a rather vague explanation in a figure caption.) Given that this is a central claim in the abstract, I can not recommend publication of the paper without substantial revision.

Reviewer #3 (Remarks to the Author):

In the manuscript entitled "Ultrasensitive nano-optomechanical force sensors at dilution temperatures", Fogliano and colleagues have shown how to optically probe the vibrations of a silicon carbide nanowire, while attaining thermalization temperatures well below 100 mK. The key aspect of the experiment is the mitigation of the heating due to optical absorption. This is achieved by implementing a detection technique based on photon counting, which in turn allows to use extremely small optical power for the probe laser, in the sub-pW regime. A combination of low effective mass, high quality factor and thermalization to temperatures below 100 mK enables a force sensitivity of $40 \text{ zN/Hz}^{0.5}$, a record value for systems with scanning probe capabilities.

The results reported by the authors are of great novelty and interest: so far, probing mechanical

motion by optical means has been limited to temperature at around 100 mK, due to the residual absorption in the bulk of the mechanical element, as the authors pointed out. The methodology developed by the authors enables reaching thermalization temperatures below 100 mK, which represents an appealing regime for very different experiments, ranging from mechanical-based sensors to fundamental studies, as testing collapse models predictions.

However, before suggesting the publication of the manuscript in the journal, I would like to raise few points and ask some questions to the authors.

1. A large fraction of the manuscript deals with the characterization of the mechanical properties in the low temperature regime, as well as a study on the heating mechanism due to optical absorption. Indeed, this is just an example of the physics one can study at these low temperature. In order to catch a larger audience and make the manuscript more visible, I would suggest the authors to rephrase the title and the abstract of the manuscript in order to englobe the thorough thermal characterization, which anyway represents a good part of the manuscript.

2. According to the authors, one of the key ingredient to achieve such low temperature while maintaining high signal-to-noise ratio in the measurements is the photon counting detection scheme. This is in contrast with the standard photodetection usually employed in continuous displacement sensors. How does the new scheme compare with other known schemes, e.g. homodyne and heterodyne detection?

3. Could the author label the equations in the Supplementary Information with sequential numbers? Readability would definitely benefit from that.

4. Some additional recent references for

(i) low thermalization temperatures attained with probe rather than optical one: X. Zhou et al., Phys. Rev. Applied 12, 044066 (2019) DOI:10.1103/PhysRevApplied.12.044066

(ii) zeptoNewton force sensitivity for scanning probe technique: M. de Wit et al., Review of Scientific Instruments 90, 015112 (2019), DOI:10.1063/1.5066618

5. The optical measurement is performed on the light back-scattered by the nanowire into the input fiber.

(i) How much is the efficiency with which the back-scattered light is re-coupled into the fiber?

(ii) Regarding the back-scattered field, is it a Gaussian or a dipole-like field?

6. The authors refer to their minimum level of optical flux used as the regime in which a single photon is collected per mechanical oscillation cycle. Is this a useful/particular regime in any sense?

7. At the end of page 4/beginning of page 5, the authors describe their fitting routine to estimate the noise temperature. I would like to have additional details and clarification about "testing a +-10% variation of the damping rate around the best fit value". At the moment, the meaning of this sentence remains obscure to me.

8. In the second column of page 5, the authors claim that "The temperature dependence of the mechanical frequency is found to be a very good indicator of the nanowire internal temperature...".

Do the authors have additional data/reference to support this claim? Can they exclude any other mechanism to generate a similar change in the mechanical frequency?

9. In the first column of page 6, around the end of the paragraph, the authors attribute the observed additional heating when probing the extremity of the nanowire to a wave guiding effect. Is this an hypothesis made by the authors? Can they back it up with a reference to existing literature? Could additional defects present at the nanowire tip explain the observed enhanced absorption?

10. At the beginning of page 7, the authors claim that "the absence of significant mechanical frequency shifts for increasing actuations strength suggests that the actuation mechanism does not heat up the nanowire.". However, before on page 5, second column, they say that the mechanical frequency shift is a good indicator for the internal temperature, and not for the mode temperature, which can be increased by parasitic force noise with negligible heat dissipation. How do the two statements reconcile together?

11. Regarding the section about force sensing: experimentally, the authors resonantly drive one of the nanowire's eigenmode (with a relative large force of $\sim 2\text{aN}$), while they monitor the displacement of the other eigenmode, thus calculate the displacement spectral density to estimate the mode temperature. From this measurement, they estimate the force sensitivity of the driven eigenmode, that is they assume that both eigenmodes have the same mode temperature. While the assumption is reasonable, the method seems a bit involved and I do not see the advantages compared to a more direct approach, in which the noise thermometry and the external force is applied on the same eigenmode. Could the author elaborate on this point? I think that a more convincing experiment for showing the force sensitivity would be to apply the smallest test force detectable in the displacement spectrum, with a signal-to-noise of around unity.

REVIEWER COMMENTS

Reviewer #1 (Remarks to the Author):

The manuscript describes experimental results on vibrations of suspended SiC NWs excited by low optical power in a dilution refrigerator. The displacement of the NWs extremity was used as readout using numerical aperture optics. The results showed a force resolution of $40 \text{ zN/Hz}^{1/2}$ and a force sensitivity in the range of fN/m . I have no doubt that observation of the oscillator thermal noise at below 100 mK temperature could be beneficial to investigating the intrinsic mechanical and physical properties of the material and structure. Also, I have no criticisms on the experimental data.

We thank the referee for his feedback on our work. Please find below the response to the comments.

It is not surprising that a high force sensitivity is observed at a low temperature of sub 100 mK. Even though this is a claim of record sensitivity for mechanical scanning force probes, I could not see where the results of this work stand in the field. Perhaps, systematic comparison of the results to the literature will help if advances have been made by this work.

In term of force sensitivity, the value of 40 zN/sqrt(Hz) represents a clear improvement (larger than 2) compared to other scanning force apparatus. To our best knowledge, the smallest force sensitivities reported for scanning force probe are:

De Wit2019: $250 \text{ zN/Hz}^{1/2}$ @ 20mK

Tao 2016: $540 \text{ zN/Hz}^{1/2}$ @100mK

Heritier2018 $94 \text{ zN/Hz}^{1/2}$ @110 mK.

Mamin2001 $410 \text{ zN/Hz}^{1/2}$ @100mK

Integrated nanomechanical oscillators, such as suspended carbon nanotubes, or single ions have demonstrated a far better sensitivity ($1 \text{ zN/Hz}^{1/2}$ in Moser2014), however this force sensitivity can hardly be used in practice since these systems don't allow for scanning force capability, in general because those are integrated planar oscillators, or because approaching a sample in a particle trap destroys the trapping mechanism.

Another figure of merit for these experiments is given by the thermal noise temperature. Our value of 32mK is on par with the best values obtained with electrical/microwave readouts ($\leq 20 \text{ mK}$ in deWit2019), and represents a significant improvement compared to other nanomechanical oscillators under optical readout.

As a remark, the difficulty to thermalize and observe nanowires at low temperatures can be fully appreciated in Fig 3d: to reduce the noise temperature from 300 mK to 30 mK, one has to lower the optical power by more than 4 orders of magnitude.

Then, for what concerns the sensitivity to force gradients, (approx. 1 fN/m for an integration time equal to the inverse damping rate) the literature is very sparse on the frequency stability of the nano-resonators, so the comparison is delicate. But since we operate with ultra-soft nanowires ($\mu\text{N/m}$), presenting an exceptional frequency stability, we believe there are not many systems that could favorably compare to the value measured in the manuscript, especially without being impacted by the probe back action as we verified numerically.

Specific work in our group will be dedicated to the investigation of ultra-weak force field gradients at dilution temperatures.

Besides experimental capability of testing at sub 100 mK, I am not really happy about the scientific aspect of this work. Understanding the phenomenon behind is important and interesting, but being treated unequally. In addition to the size of the nanowire, material is the key factor determining the response of the optomechanical force sensor.

This paper represents the first demonstration of the capacity to operate with such a force sensitivity with nano-optomechanical nanowires at dilution temperature.

This type of ultra-sensitive experiments is demanding and rare (see previous point). The demonstration of thermalization down to 32 mK is major step towards our long term goals (single photon optomechanics and hybrid spin-nanomechanical interaction).

The next part of our work will be to make use of this high sensitivity setup to explore more systematically thermal and mechanical properties of the nanowires at low temperature, leading naturally to the study of material properties in an original geometry. We discuss this aspect in the following.

The mechanical properties of SiNWs were observed as large similarities with those in amorphous materials (figure 4). Did the author confirm if SiC NWs was amorphous? In some discussion, the authors mentioned cubic silicon carbide (3C). Are there any differences in the behaviour of the mechanical system if there are a change in material structures?

As explained here, even if our SiC nano-resonators present similar mechanical quality (stiffness, damping rate) for different allotropic phases, generally the mechanical properties of resonators significantly vary with the amorphous/crystalline character of the material.

The nanowires employed in this work are dominantly crystalline, of 3C phase, CVD grown. But they can present a lateral oxide crust and variations of the allotropic phase along their growth axis, as revealed in TEM measurements (and variations of the allotropic phase as illustrated). Here follows an image taken at ILM Lyon of SiC nanowires. The nanowires we employed are of the 3C phase, and we select them optically to avoid allotropic defects (which can be detected through color changes under white light illumination) thus the hexagonal phases are not present, but the oxide crust remains.

Figure 2. a) Conventional TEM image of a typical SiC NW exhibiting an SF area (dark part) and a pure 3C-SiC section (bright part). b) High-resolution image of the interface between those two crystallographic zones. This image clearly shows the random network of SFs and the pure 3C-SiC component.

TEM Image taken from Bechelany et al, *Adv. Funct. Mater.* 17 939 (2007).

In our room temperature experiments, we also employed hexagonal types, which are of identical mechanical quality, but present far more allotropic defects along their growth axis. This is annoying when using optical readout at dilution temperatures, since the fringe pattern will strongly vary with the vertical position of the nanowire within the optical mode. The interest of hexagonal types (4H, 6H) is that they present a higher optical bandgap compared to the 3C phase, which can be interesting for an optical readout in the visible.

We cited the reference Bechelany et al, *Adv. Funct. Mater.* 17 939 (2007) in the main text when referring to TEM imaging (p5).

Coming back on the referee question concerning the amorphous nanoresonators:

Operating with amorphous resonators is often seen as a source of problem for ultrasensitive force sensing: the materials are not extremely stable, they can present a creep, causing long term logarithmic frequency shifts, which is not desired for gradient force measurements. Then their mechanical quality factor are in general relatively limited compared to crystalline materials, even if this discrepancy tends to be reduced at ultra-low temperatures, once the TLS governed dissipation enters the tunnelling regime, below 0.1-10 K typically depending on the frequency (see Enss2005).

As described in the main text (p5), the temperature dependence of nanowire mechanical properties presents similarities with amorphous materials, but the differences (higher quality

factors, smaller frequency shifts) are also underlined and the mechanical properties of resonators significantly vary with the amorphous/crystalline character of the material.

Taking into account the referee remark we added in the main text (p5) :

« Although the SiC NW are dominantly crystalline (3C phase), the temperature dependence of the mechanical properties presents large similarities with ... amorphous material

From engineering point of view, SiC NWs are selected but crucial material characteristics that are beneficial for optomechanical force sensors could be discussed. Material structures ranging from amorphous, nanocrystalline, polycrystalline and single crystalline with different doping levels could affect the mechanical and electrical properties, as well as the interaction between optical input and NWs structures, especially at low temperatures.

For the moment we have not yet gone through a very large screening of samples made of different materials, mainly because the results obtained with SiC nanowires were in line with our long term objectives (single photon optomechanics and hybrid spin-nanomechanical interaction) but in principle the readout techniques employed can be applied to many other nanomechanical systems. In our experiments, we have tested silicon carbide nanowires, GaN nanowires, SiN doubly clamped nanowires, metallic nanowires, and aluminium membranes, and realized preliminary tests with doubly clamped carbon nanotubes.

For what concerns the mechanical dissipation, we believe it is reasonable to aim at operating with purely crystalline materials, in order to avoid internal dissipation which is usually larger in amorphous or polycrystalline samples, especially at low temperature.

Since we operate with nano-resonators with ultra-low thermal conductance, it is very important to minimize the light absorption in the system. This is why we work in general with large bandgap materials such as silicon carbide, GaN or diamond nanowires, and we have avoided Silicon or GaAs nanowires, even if the material processing techniques are much more advanced for those nano-systems. Then, the optical absorption can be increased by surface defects or contaminants, and there the sample preparation plays an important role.

The doping level can clearly play an interesting role in the future, especially if it can increase the heat conduction without affecting to much their mechanical dissipation.

Then, the optical contrast could be increased by nano-engineering the nanowire surface, to increase the optical reflection efficiency by patterning a 1D photonic crystal. The overall reflection efficiency is of a few percent, and any increase would help reducing the amount of light needed to probe their thermal noise. However this has to be done without increasing the overall light absorption of the nanowire.

Finally, the precise estimation of the thermal conductance of the nanomechanical resonators is delicate without direct measurements. This represents a research domain in itself. We operate in principle in a regime where it becomes ballistic, but the measurements are difficult and rare, so this can be delicate to anticipate. However, measurements of the optical heating rate as described in the manuscript represents a novel methodology which will help exploring those questions in the future.

As thermalized down to sub 100 mK, noise temperature from unwanted heating could be

significant. External drives could increase the stiffness and vibration frequency, while thermal heating has an adverse effect. ...

We are afraid we do not completely follow the referee comment. We assume the referee discusses the analysis of the frequency/damping rate as a function of temperature shown in Fig. 4.

First we have worked to mitigate the electrostatic and mechanical noise sources, and have developed optical measurement techniques which allow to drastically limit the optical absorption to an almost negligible level. At the end we could approach a noise temperature level of 32 mK, close to the sample temperature 27 mK measured with a standard thermometer.

After a proper mitigation of external source of noise (electrostatic and acoustic), optical absorption is the leading heating mechanism which causes an increase of the nanowire internal temperature.

The remaining temperature difference should originate from uncompensated noise sources, or from a residual optical heating.

As a remark, as we verified experimentally (Fig. 5 and text), our force probes are operated in the linear regime, which extends up to large oscillation amplitudes (10 nm), for which a large signal to noise of approx. 40 dB is obtained, without leading to appreciable heating of the nanowires. Furthermore, as discussed, the ultra-low dissipation of the force probe leads to a negligible power dissipated by internal friction, compared to the optical powers involved. As such, the remaining artificial drive will not affect the nanowire noise temperature.

But they possibly can, as the referee suggested, modify the nanowire stiffness through a quadratic AC electric noise for example (the nanowire frequency depends quadratically on the electric field (capacitor like effect), and thus any AC noise could generate a frequency shift proportional to its rms value).

Then, concerning the evolution of mechanical properties with temperature:

The first measurement we realised is done by increasing the cryostat temperature while keeping an ultra-low optical power: there, it is possible that the noise environment evolves with the temperature: the acoustic noise in particular can be modified due to the modification of the circulation conditions of the helium mixture, displacement of the helium 3/ mixture interface, change of evaporation rates in the still, etc.

We note however that due to the architecture of the thermal heater, the sample temperature increases 4 times more than the mixing chamber temperature, so that when we increase the sample temperature up to 400 mK, the mixing chamber only rises to 100 mK.

The second measurement is realized by keeping the cryostat cold, and progressively increasing the optical power. The cooling power of the cryostat is very large ($6\mu\text{W}$) compared to the optical powers injected, so that the operating point of the cryostat remains quasi unchanged (within $100\mu\text{K}$), and as such the acoustic noise properties should not be modified. In that sense, the mechanical frequency increase observed at low optical power cannot be explained by changes in the artificial external noises.

Then, the optomechanical effects only start playing a role at larger optical powers, above $1\mu\text{W}$, while we only employed 10 nW at maximum in Fig. 4.

Based on our past experiment, we usually use a $1\text{fN}/\mu\text{W}$ scaling factor for the optical force, and since the optical forces vary on half the optical waist dimensions ($500\text{ nm}/2$), the force gradient expected scales as $\text{dxF} \sim 4\text{e-}9\text{ N/m}/\mu\text{W}$. The later will cause a typical frequency shift of $\text{df/f} = \text{dxFx}/2\text{M}/\omega\text{m}^2 \sim 1.5\text{e-}4/\mu\text{W}$, which amounts to a maximum relative shift of $1.5\text{e-}6$ for 10 nW , if the nanowire was located on the side of the optical waist, which is not the case in the experiment. Thus the optomechanical force gradients are negligible in this measurement.

Finally, the evolution of the mechanical frequency shifts with temperature has the same shape as the one observed on amorphous materials, except that it features a slower log dependency, which is attributed that he fact that the oxide crust is rather thin compared to the nanowire diameter.

... Perhaps, location of excitation on the SiC NWs could be considered if there are considerable mechanical frequency shifts as heating has more significant impact toward the fixed end. This needs clarification.

We are currently conducting thermal response measurements realized by positioning the pump laser, intensity modulated, at different positions along the nanowire, while measuring its thermal response driven by a second laser. All the nanowires tested up to now present an increase of their thermal response when the laser is positioned in the last $1\text{-}2\ \mu\text{m}$ away from their extremity. The heating efficiency is reduced at positions above that region. This is illustrated on the following measurement.

Heating efficiency, deduced from pump-probe response measurements, for different positions of the heating laser in the vicinity of the nanowire extremity (located on the right). The enhanced absorption is attributed to a wave guiding mechanism, which increases the amount of light which is channelled into the nanowire, and is thus more likely to induce optical absorption.

In general, the thermal response remains almost homogeneous above that area, except when the laser is positioned at a position where the nanowire presents a local geometrical defect, a

local diameter change in general, where one can observe a higher absorption rate. We think that this is likely a consequence of a higher amount of light being injected inside the nanowire volume.

In order to mitigate the optical heating, we avoid operating close to the nanowire extremity, or close to geometrical defects, and generally position the readout laser 2-5 μm above the vibrating tip. This has a minor impact on the nanowire effective mass, in view of their pretty large length (>200 μm).

Some minor points are as follows. Please proofread the manuscript, some typos are detected, e.g. 700 C; make clear references to the used equations; to make sure experiments are well controlled, discussions should be given on selection of vibrating frequency ranges and quality factor; information about material synthesis would be informative.

We have done our best to suppress the remaining typos. An abacus is given in the SI to target the desired nanowire geometry (length/diameter), which may vary from one application to another.

Following the referee suggestions, we have added a paragraph in the SI describing the nanowire preparation in greater details.

Overall, the authors present nice experiments and results. I advise publication of this work if the above points have been addressed. It is interesting to prove and clarify the science behind achieving ultra-high sensitivity of optomechanical force sensors. I would expect better guidance for researchers in design system for ultrasensitive force measurements.

We thank the referee for his comments and hope we have addressed the points he raised. We believe the corresponding modifications of the article will give a better guidance to researchers in the community

Reviewer #2 (Remarks to the Author):

This work describes the characterization of SiC nanowire probes at millikelvin temperatures for eventual use as force and force gradient sensors in scanning probe applications. Because the optical readouts used for these applications usually generate more heat than a dilution refrigerator can dissipate, the authors propose a single photon interferometric readout scheme that allows them to drop their photon fluxes below about $3\text{E}6/\text{s}$ (~ 1 pW). They use this capability to measure the mechanical properties of the nanowire cantilevers as a function of temperature, establishing a consistent value for cantilever effective mass regardless of whether the nanowire was heated by changing the temperature of the cryostat or by changing the photon flux. The latter technique also allows them to make some general statements about the thermal conductivity of the SiC nanowire. The optical properties of the nanowire are potentially obscured by the surface contaminant layer. They then make some claims

about the sensor's force and force gradient resolution in conclusion.

In general, this is very good work. The experiment was challenging, and the authors appear to have conducted it with attention to detail in both the lab and in the theoretical approach. There is clear evidence of interference fringes in Figure 1, and although a similar approach has been used previously to extract the interferometer sensitivity, it has not been done with single photon detection. The characterization of the material properties of the nanowire itself are likely to have less impact, as the authors acknowledge the presence of an oxide layer and surface contaminants make the results not straightforward to interpret. In addition, several points about the claimed force resolution must be clarified before the work is ready for publication.

We thank the referee for his feedback on our work, please find below our response to the comments.

Specific Comments:

Page 1, Right Column, Paragraph 3: The use of the phrase “cavity-free” is not entirely justified. The authors have created a low-finesse optical cavity with the fiber face as one end and the nanowire as the other end. A cavity of some kind is necessary for this type of homodyne interferometer. It would be more accurate to say this system is free of a resonant cavity.

We agree with the referee that in principle one cannot fully neglect cavity effects. However they are very low here for the following reasons:

First, the reflectivity of the fiber extremity is around 4%, and the reflexion efficiency from the nanowire to the fiber mode is of the same order of magnitude, so the maximum finesse of the cavity would be around $2\pi/(2*(1-0.04))=3.2$. Furthermore we often play with the light polarisation to enhance the interference contrast, using the fact that the reflexion on the fiber is polarisation insensitive, while the nanowire introduces some birefringence in reflexion (please refer to our reflectivity coefficient plots in the SI). In such a situation, the field reflected from the fiber and the nanowire will not present the same polarisation state, meaning that this will reduce even more the effective cavity finesse, and the interference only reappear at the polarisation analysis step, realized outside of the cryostat.

Also, the optical force experienced by the nanowire (which we can measure using a pump probe scheme, Glippe2014, Fogliano2019) does not vary along the optical axis in the Rayleigh length area, suggesting that the possible cavity effects are almost negligible (if the cavity effect would have played a meaningful role, one would have expected to observe a $\lambda/2$ periodicity, (see our work on cavity nano-optomechanics: Fogliano et al arXiv:1904.01140 for example), and this is not the case here).

In view of these considerations, we find that the phrasing “non-resonant cavity” is not really adequate, and not necessary to our measurement neither. Our approach is closer to a Michelson interferometer, with both arms folded along the same fiber to minimize the drifts in the optical path difference. Also we find that the phrasing “homodyne measurement” can be misleading with respect to its traditional implementation in quantum optics since the local

oscillator has here a similar strength as the signal (homodyne detection is in general used to get rid of the detector dark noise, which is not a limitation here.)

To follow the referee comment, we have simply written “interferometric optomechanical readout”.

P2, L Column, Ppgh1: The phrase “length over diameter” is more succinctly expressed as “aspect ratio.”

We have followed the referee suggestion

Figure 1: A few of comments about this.

1. Please clearly label the coordinate system inset into Fig. 1e. It is difficult to parse Fig 1f without an ambiguous visual cue as to which direction x and z refer to. In general, z often aligns to gravity, but that would not make sense in light of the diagram in Fig. 1e and the interferogram in Fig. 1f.

The xyz coordinates are attached to the coordinates of the piezo scanner supporting the nanowire.

We generally align the optical axis with the z coordinate, as commonly employed in optics, the y axis being vertical, pointing towards the bottom direction so that the xyz frame is direct and the xz frame is positively oriented when looking from the top.

We have added a xyz diehedral in Fig 1e to clarify the orientations of the experiment.

2. I notice a PBS and two APDs in the experimental diagram of Fig 1e. Was this a Hanbury-Brown-Twiss interferometer? A brief explanation of the dual APD setup is warranted.

Depending on the type of measurements, one can use standard photodiodes at large power, or avalanche photodiodes in reflexion, and the same happens in the transmission channel. Since the nanowire reflexion is polarisation sensitive (see the SI plot), but not the reflexion on the fiber extremity, it can be interesting to record 2 reflexion signals (after splitting on a PBS) to analyse the reflected polarisation structure and optimize the readout efficiency.

To avoid confusion, we have removed the second photodiode on the figure, and added a paragraph and figure in the SI describing the optical scheme of the experiment and the polarisation control.

3. Would like some clarification about which part of the nanowire is generating the contrast. It looks like the center of the fringes in Fig.1f tail off at higher and lower values of x. The text hints that this is at the far distal end of the nanowire, but this should be clarified.

The map in Fig. 1f is obtained by scanning the nanowire vibrating extremity horizontally, in the transverse xz plane. At the waist the optical spot is sub- μm in size, so it is only a tiny fraction of the nanowire length which is illuminated. The image of fig 1f is obtained by positioning the laser 2 μm above the nanowire extremity, but such a map can be reproduced at

any vertical position, and this serves in general to inspect the nanowire homogeneity, the possible angular tilts, etc.

There can be a slight tilt of the optical axis, set by the fiber objectives, with respect to the piezo scanner coordinates. Also, the wave front can be slightly distorted (even for perfect optical elements if the fiber output and the lenses axes are not perfectly aligned) in the objective and this will be more visible away from the waist area. The use of a nanowire to image all those possible imperfections indeed helps us to align and optimize the lens/fiber arrangement in the building phase of the fiber objectives. Then the nanowire axis, may not be perfectly aligned with the scanner coordinates and the optical axis, this can cause some slight transverse asymmetries in the interference map. Finally, we operate with relatively large numerical apertures, so that the paraxial approximation is not valid: the incoming beam is not perfectly Gaussian, which is visible far from the waist, especially on the right hand side, where an internal structure appears in the map. Also the polarisation state is not homogeneous within the waist, especially on the sides of the optical waist, which can be detected and analysed using polarising elements in injection and reflection. In the article, we mainly operate on the optical axis, where those deviations from the paraxial description in the horizontal plane are minimized.

As such, the fact that the interference pattern is bended away from the optical axis is simply due to the bending of the optical wave-fronts, they are converging before the waist and diverging after.

We have added "...($2\ \mu\text{m}$ above its vibrating extremity) in the waist area, in the xz horizontal transverse plane,..." in the caption for clarity.

We have added a refined description of the fiber objectives and their preparation in the SI.

P3, L, 2: Relatively high NA objectives are used in this work. This severely restricts the working volume accessible to the experimenter, as any obstruction of the optical path will cause diffraction of the sensing laser used in the interferometer. Given that the nanowires are only a few hundred micrometers long, it will be extremely difficult to get a surface into close proximity. This also relates back to my question above about where on the nanowire the interferogram of Fig 1f is measured. Many of the fields that are of most interest to scanning probe microscopists decay rapidly with distance. If the authors want to use scanning probe microscopy as the primary justification for the relevance of their work, they should provide a realistic estimate of the spatial resolution they can expect to achieve with the sensor during operation in a scanning probe mode.

The lateral optical readout can be a problem to realize force measurements above an extended sample. However, it is possible to move the probe laser up along the nanowire and away from its vibrating extremity without losing significantly on the effective mass and thus on the signal to noise or signal to background ratio:

In view of the mode profile measured in the above plot, one loses less than 50% on the effective mass as long as the readout laser is positioned in the last 20% of the nanowire length. And the increase in effective mass is of 8.7 if the laser is positioned at half the nanowire length, which is significant if one wants to probe the nanowire thermal noise, but non-impacting if one operates with driven trajectories.

For a 200 μm long nanowire, we can then position the laser up to 40 μm above its vibrating extremity and only reducing the SNB by 3dB (in practice the SNR are around 50 dB for driven motion in our room temperature experiments, this is thus not a limitation).

This then allows us to measure forces on spatially elongated samples, extended perpendicularly to the optical axis, and when it is not possible, we work on the side of a half planar sample, up to 40 μm from the edge, without cutting the incoming/reflected light fields (for a NA of 0.7, the total focussing angle is around 90°). All our force sensing scanning probe experiments are realized as exposed.

What can be more delicate is the fact that when we scan a sample below the nanowire, to realize scanning probe measurements, part of the light can be back scattered by the sample into the optical readout channel in reflexion, which can modify the measurement vector. To compensate for this problem, we have implemented measurements protocols which permanently measure

the measurement vector (e_{β}) in real time. To do so, we dynamically move the nanowire along both transverse directions, using to coherent drives (80, 85 Hz) and demodulate the measurement channel at each frequency to deduce the local measurement vector.

For what concerns the spatial resolution of the force sensor: it can be limited by the thermal noise of the probe, by the geometry of its extremity, and by the amplitude of oscillation employed when one wants to use the dynamical force sensing protocols. At dilution temperatures, the thermal noise spreads over $\sqrt{k_B T_{\text{eff}} / M_{\text{eff}} \Omega_m^2} \sim 0.2$ nm rms,

while the NW diameters are around 100-300 nm. For very local forces, such as Casimir/Van der Waals forces, sharpening the tip extremity thus helps increasing the lateral resolution of the nanowires, but this could be less efficient for electrostatic forces, which decay less rapidly with the distance to the surface to explore.

At room temperature, we employ oscillation amplitudes around 10 nm, while at low temperatures, the signal to noise is already extremely large with 1 nm. In the scanning force sensing measurements we realize at room temperature, we can observe lateral details of a few tens of nm, but this does not seem to be an intrinsic limitation.

We have added the figure on the longitudinal spatial profile in the SI and discussed the laser positioning along the nanowire in the “optical heating” section:

“In view of the relatively long nanowires employed, we can position the readout laser at the least absorbing location in an extended area close to their vibrating extremity without suffering from a significant increase of the effective mass (see SI).”

Figure 3: Figure 3a, are the two peaks in this subfigure intended to represent two different vibration modes? If so, the color coding should match the inset and be consistent with figure 3 c and d.

The color code is the same as in Fig. 2c, the 2 peaks represent the 2 transverse fundamental modes. In the manuscript, red is the color for the low frequency mode, and blue for the higher frequency mode. The color codes are consistent with the insets, which are zooms of the noise spectra. Fig. 2c, 3a, 3b are consistent in the color code.

Fig 3c represents noise thermometry realized on the higher frequency mode, so the blue color is adapted (the red point was meant to indicate the noise temperature measured in absence of suspension mechanism).

The heating curve shown in Fig 3d represents 2 different measurements, on the same mode so they should be both colored in blue-like colors.

We have changed the red dot in Fig 3c into a dashed line for clarity (it only represents a typical noise level, thousands of measurements were realized prior to reducing the acoustic vibration noise with the suspension apparatus).

We have modified the color code of Fig. 3d so that the reader does not get confused.

P5, L, 1: Please clearly explain the distinction between nanowire temperature and sample temperature.

The “effective temperature of the nanowire position noise” refers to the temperature derived from the thermal noise thermometry, while the “sample temperature” is measured on a RuO₂ thermometer connected to the sample support. In the manuscript, we also evoke the cryostat base temperature, which is also measured with a physical thermometer (RuO₂+carbon+Pt100 resistors) placed on the mixing chamber, the coldest spot in the cryostat. For the measurement discussed above, the base temperature was of 22 mK, but the noise temperature should be compared to the “sample temperature” (27 mK).

For clarity, we have specified the thermometer type, and given the mixing chamber temperature.

For clarity, we have homogenized the name of the vibration noise temperature all across the manuscript: T_{eff}^m is the noise temperature of each mode (m being the mode number), and we may use T_{eff} , when there is no ambiguity.

Figure 4, in Fig. 4a, presumably the two colors are meant to represent the two transverse vibrational modes. Optional suggestion: make different shapes for clarity in case the information is displayed in a format without color.

Yes, here as well the red/blue color code represents the lower/higher frequency fundamental eigenmodes.

We have changed the shape of the lower frequency mode dataset.

P6, L, 2: There is a lot of information crammed into this paragraph. From briefly reading reference 39, I think that the statement “a temperature can be defined quasi-continuously along the nanowire” needs to be clarified. Is this intended to state that the temperature gradient is nonzero and continuous along the length of the cantilever, or that the cantilever is at a constant uniform temperature along its length? Explain more how this contrasts with the case above 150 mK? Although this may be explained more fully in the supplement, a better explanation in the body of the paper is necessary.

When the phonon mean free path is larger than the nanowire length, it is not possible to correctly define a temperature within the nanowire since the phonons cross it without interacting so that no equilibrium can be reached. When one follows the existing literature, this is expected to occur below 150 mK, meaning that below this temperature, all the discussions on the nanowire thermal conductance are in principle not valid. However, one must be cautious with this temperature threshold estimation: it largely depends when the transition from the Casimir regime to the Ziman regime appears, which depends on the nanowire effective surface rugosity, a measurement which is not easy and only estimated in our case, and will generally depend on the precise sample geometry (local constrictions due to a change in the allotropic phase of the nanowire, surface crust, etc).

In the following experiments, we will aim at investigating in greater detail how the heat propagates along the nanowire, using pump-probe measurements with a second movable heating laser spot (as sketched in the image shown in our response to referee #1).

Then, if one assumes that the above criteria is met, so that the temperature can be defined continuously along the nanowire, the laser heating will generate a temperature gradient along the nanowire, which will not be homogeneous (linear temperature profile) if the nanowire local conductance depends on the temperature. Since the latter always increases with temperature, and since the vibrating extremity of the nanowire will be warmer, the heat will be carried out more efficiently close to the laser compared to close to the supporting tip, where the temperature should remain close to the sample temperature. As such the temperature gradient will be smaller there compared to the upper area. As a consequence, in presence of a large variations of the nanowire local conductivity with temperature, the temperature profile inside the nanowire will be almost homogeneous, and all the temperature gradients will be “pushed” towards the clamping area. The model given in the SI explains this mechanism in greater detail.

We have clarified the wording of the paragraph:

“ In the limit where the phonon mean free path remains small compared to the nanowire length, which occurs in principle **only** above 150\,mK, see SI, a temperature can be defined quasi-continuously along the **nanowire (otherwise the phonons may travel along without interacting)**. Then, when the conductance strongly increases with temperature, the temperature gradients in the nanowire become largely localized close to the supporting tip, which is assumed to be thermalized to the cryostat. The internal temperature profile is then expected to be almost homogeneous within the vibrating part of the nanowire and in first approximation, can be assimilated to the measured thermal noise temperature. **This remark explains the similar behavior** observed between both measurements in Fig.\,4.”

P6, R,1: If the optical absorption is dominated by contaminants, what is to say that the thermal conductivity is not also? The arguments addressed in the previous comment (P6, L, 2) fall apart if there is something with much higher thermal conductivity coating the outside of the cantilever.

Our optical heating measurement (such as in Fig 3d, 5d) allows us to determine the absorption/conductance ratio of the nanowire, however none of them is known at such low temperatures for the nanowire employed. If we assume that at high temperature (10K), the thermal behavior of the nanowire is well described by the Casimir model, then we obtain a 70 ppm absorption ratio, which we find rather large for a high bandgap material. We have a rather limited knowledge of the material properties of the nanowires so we can only speculate on the origin of this absorption. TEM measurements have shown (see the corresponding image in our response to referee #1) that they also present a thin oxide crust on top of a crystalline core, so the excess of absorption can be due either to internal defects / dopants in the crystal, or to surface absorption either due to the oxide crust itself, or to surface contaminants.

The conduction through the oxide crust may play a role, but it is in principle far less efficient than the one of the bulk crystal at low temperatures (if one adopts a naïve geometrical model for the conductance: a cylinder of SiC surrounded by an oxide crust): first its cross section (typically 1-5 nm in thickness) is much smaller than the nanowire SiC internal surface, second the material conductivity is rather small compared to SiC which remains a very good heat carrier even at low temperatures.

Using existing models for bulk materials, we can estimate the conductance of the oxide layer (PhD thesis Adib. Tavakoli, UGA 2017, Enss2005). We found that the amorphous crust (3nm) would represent a conductance around 20% of the Casimir conductance at large temperature (10K). Since the amorphous conductance rapidly decays with temperature (T^2) it becomes rapidly hidden by the Ziman conductance (T^1) and should not play any dominant role in our nanowires at very low temperatures. See the modified figure below.

It is however worth noting that the oxide crust is likely to increase the scattering rate of phonons on the nanowire boundary, and thus plays a role in the transition from the Casimir to

the Ziman regimes. Future investigations, aiming at removing this oxide layer, will help responding to that question.

Figure on the conductance analysis in the SI: the TLS contribution has been added.

For what concerns the absorption inside the crystal core, it could be due to a residual contaminant or doping, which could clearly play a role in the thermal conduction properties of the nanowire, especially if electrons start to have a meaningful contribution. In the future we will try to obtain more information on that aspect, which is for the moment not under control in our samples. We note however that our measurements of electrostatic forces and work by colleagues in Institut Lumière Matière in Lyon, permit to estimate their electrical resistance on the order of a few tens of GOhms, which brings a resistivity around 1 Ohm.m at room temperature. This order of magnitude, even if it exceeds the one of pure insulators, remains very small compared to metal or doped semiconductors, and it is believed to be governed by surfacic conduction channels.

Finally, we can measure the thermal cut-off of the nanowires, using pump-probe techniques, from which we can deduce their thermal conductivity. Above this cut-off frequency, the thermal wave does not have time to diffuse up to the clamping point. Our measurements at room temperature allow us to measure the cut-off frequency around 80 Hz for the 3.95 kHz nanowire, a value which matches well with the cut-off value derived from the bulk parameters. The objective is now to reproduce those measurements at low temperatures, and this should help us understand more their thermal conduction properties.

We have added the discussion on the conductivity of the oxide crust in the SI.

Fig 5. In figure 5b, the minimum driven amplitude (RMS, as I assume this is measured with a lockin amplifier, given the units) is approximately 1 nm. From the mass and first resonant

frequency, I deduce that the stiffness of the relevant vibrational mode of the nanowire is $2.5E-6$. At resonance, the response of the nanowire cantilever is enhanced approximately by a factor of Q relative to the quasistatic case. I deduce Q to be approximately $1e5$ from the data in figure 5b, which leads me to conclude the minimum force measured by nanowire A is approximately 200 zN based on the data available.

The measurements are indeed realized with a lock in, but we have plotted the amplitude value (not rms), and the parameters employed are the following:

What is plotted is the oscillation *amplitude* (not rms), and the minimal driven oscillation amplitude is 0.6 nm. The mechanical linewidth is 40 mHz, corresponding to a quality factor of 98000, while the effective mass is $16e-15$ kg.

As such the static susceptibility of the nanowire is

$$\chi_0 = \frac{1}{M_{\text{eff}}\Omega_m^2} = 1.0 \text{ e5 m/N},$$

corresponding to an effective stiffness of $9.9e-6$ N/m, while its resonant mechanical susceptibility is

$$\chi_{\text{res}} = \frac{1}{M_{\text{eff}}\Omega_m\Gamma_m} = 1.0e10 \text{ m/N}$$

Thus the smallest driving force amplitude (not rms) is

$$0.6 \text{ nm} / \chi_{\text{res}} = 59 \text{ zN},$$

which is the minimal drive strength stated in the manuscript (the measurements are realized with a 50mHz resolution bandwidth). The force sensitivity of this nanowire is $\sqrt{2 M_{\text{eff}}\Gamma_m k_B T_{\text{eff}}} = 59 \text{ zN/Hz}^{0.5}$, in agreement with the value expected from the abacus (after correction of the effective temperature: 32 vs 20 mK) given in the SI.

We note that the force magnitude which is stated in the manuscript implicitly corresponds to a dynamical force presenting a spatial profile identical to the optical readout profile (which serves to determine the effective mass of the oscillator, see Pinard et al, EPJD 7 107 (1999)). In our case, since the optical waist is very small compared to the nanowire length, it corresponds to a point-like force localized at the nanowire extremity. This is pertinent when investigating the optomechanical force exerted by the readout beam for example, as largely explored in the group (Gloppe 2014, Mercier de Lépinay 2017, Fogliano 2019), but also for a scanning probe operation since the forces under investigation are in general rapidly decaying with the distance to the sample under test.

In that sense, one should employ the effective stiffness and not the static stiffness (4 times smaller, which explains, in addition to the rms aspect, the difference between the referee estimation and our value).

We have added the mechanical linewidth in the caption, which was missing there, and explicitly stated that this was not a rms amplitude. We have also clarified the theoretical framework associated to the force magnitude estimations, and added a reference to Pinard et al., where the formalism originates. We note that when one employs the Langevin force noise expression ($S_F = 2 M_{\text{eff}} \Gamma_m k_B T_{\text{eff}}$) to estimate the minimum force sensitivity, those considerations on the force geometry are always implicitly used.

Following the referee and referee #3 comments, we have also updated the plot, to make the background level appear.

The effective mass of nanowire C is not reported.

The effective mass of nanowire C is 1.6pg, we have added it in the manuscript. Please note that it is not a straight cylinder, but a conical nanowire, so it does not fit directly in the abacus discussion found in the SI.

The information justification in the text for the putative 40 zN/Hz¹² noise floor is not clear. According to the supplement, the measurement stage was not suspended, and there appears to have been some background subtraction that is not adequately described in the text.

The background subtraction mentioned in the SI is not related to the stated force sensitivity:

We understand that the referee refers to the sentence “The dephasing induced by the piezo actuation chain has not been subtracted” in the caption of Fig SI 6.

If yes: when we drive the nanowire with a piezo or with any other actuation source, there can be a delay between the output voltage modulation, produced by the lock-in, and the time-modulated force experienced by the nanowire, due to the entire actuation chain: amplifiers/filters, internal resonances in the driving piezo. As a consequence, the phase response is delayed by a fixed quantity (due to the very narrow frequency span employed, the correction is identical for all frequencies), which we have not subtracted here as stated.

But this remark only concerns the phase plot of the response measurement, and do not impact our claims on the force noise level, which are related to the thermal noise of the second mode (non-driven) of the nanowire.

Then, the sentence concerning the role of the suspension apparatus, meant that even in absence of suspension, we could observe some rather cold noise temperatures. This actually depends on the frequency of the nanowires since the artificial excess of noise is not the same at all frequencies. In particular, this colored noise is found negligible at 11 kHz where noise temperatures around 50 mK could be measured in absence of suspension, while it had a large impact at 3.9 kHz. This is not really surprising, since the cryogenic experiment presents some internal mechanical resonances and the channelling of the vibrations to the experimental plate is expected to largely vary with frequency.

The appropriate method for judging claims of minimum force noise in published literature (especially in high impact journals) is that the authors clearly provide an easily understandable justification for their claims without reference to the supplementary materials (which, in this case, are a rather vague explanation in a figure caption.) Given that this is a central claim in the abstract, I can not recommend publication of the paper without substantial revision.

We agree with the referee that the main text was too brief when explaining the force sensitivity.

The methodology which is followed for nanowire C is similar to the one employed with nanowire A, which is described in the manuscript. As commonly done in the literature. We deduce the noise temperature from a fit of the projected thermal motion spectra, using the equation given in the manuscript of $S_{\delta r_{\beta}}$, which allows us to determine the noise temperature of the mode, and conclude on the limitation on the force sensitivity cause by the thermal noise of the nanowire. The latter is given by the noise spectral density of the Langevin force, given by

$$S_F = 2 M_{\text{eff}} \Gamma_m k_B T_{\text{eff}}$$

Here M_{eff} is the effective mass (1.6 pg), determined at larger temperatures, where there is negligible optical heating, Γ_m and T_{eff} are determined from the fit of the thermal noise spectra (120 mHz and 47 mK). Those numbers gives the minimum force sensitivity of $\sqrt{S_F} = 40 \text{ zN/Hz}^{1/2}$ as stated in the manuscript.

The figure in the SI was initially meant to give complementary information to the measurement shown in Fig 5cde, but we agree that the text was too quick on the aspect of the force sensitivity. We have modified the manuscript accordingly, and in particular given the expression of the driven response, and the one of the limiting force noise. We have added a table in the SI listing the properties of the nanowire employed, including the lowest noise temperature measured and the limiting force sensitivity.

We believe the explanation of the force sensitivity is now clarified and the main text allow to appreciate the very weak minimum force sensitivity.

Reviewer #3 (Remarks to the Author):

In the manuscript entitled "Ultrasensitive nano-optomechanical force sensors at dilution temperatures", Fogliano and colleagues have shown how to optically probe the vibrations of a silicon carbide nanowire, while attaining thermalization temperatures well below 100 mK. The key aspect of the experiment is the mitigation of the heating due to optical absorption. This is achieved by implementing a detection technique based on photon counting, which in turn allows to use extremely small optical power for the probe laser, in the sub-pW regime. A combination of low effective mass, high quality factor and thermalization to temperatures below 100 mK enables a force sensitivity of $40 \text{ zN/Hz}^{0.5}$, a record value for systems with scanning probe capabilities.

The results reported by the authors are of great novelty and interest: so far, probing mechanical motion by optical means has been limited to temperature at around 100 mK, due to the residual absorption in the bulk of the mechanical element, as the authors pointed out. The methodology developed by the authors enables reaching thermalization temperatures below 100 mK, which represents an appealing regime for very different experiments, ranging from mechanical-based sensors to fundamental studies, as testing collapse models predictions.

However, before suggesting the publication of the manuscript in the journal, I would like to raise few points and ask some questions to the authors.

1. A large fraction of the manuscript deals with the characterization of the mechanical properties in the low temperature regime, as well as a study on the heating mechanism due to optical absorption. Indeed, this is just an example of the physics one can study at these low temperature. In order to catch a larger audience and make the manuscript more visible, I would suggest the authors to rephrase the title and the abstract of the manuscript in order to englobe the thorough thermal characterization, which anyway represents a good part of the manuscript.

We thank the referee for his feedback on our work.

We agree with the referee and we modified the title and the abstract to take into account the above mentioned aspects:

“Ultrasensitive nano-optomechanical force sensors operated at dilution temperatures”,

Cooling down nanomechanical force probes is a generic strategy to enhance their sensitivities through the concomitant reduction of their thermal noise and mechanical damping rates. However, heat conduction becomes less efficient at low temperatures, which renders difficult to ensure and verify their proper thermalization. We implement optomechanical readout techniques operating in the photon counting regime to probe the dynamics of suspended silicon carbide nanowires in a dilution refrigerator. Readout of their vibrations is realized with sub-picowatt optical powers, in a regime where less than one photon is collected per oscillation period. We demonstrate their thermalization down to $32 \pm 2 \text{ mK}$, reaching record sensitivities for scanning probe force sensors, $40 \text{ zN/Hz}^{1/2}$, with a sensitivity to lateral force field gradients in the fN/m range. This opens the road toward explorations of the mechanical and thermal conduction properties of nanoresonators at

minimal excitation level, and to nanomechanical vectorial imaging of faint forces at dilution temperatures.

2. According to the authors, one of the key ingredients to achieve such low temperature while maintaining high signal-to-noise ratio in the measurements is the photon counting detection scheme. This is in contrast with the standard photodetection usually employed in continuous displacement sensors. How does the new scheme compare with other known schemes, e.g. homodyne and heterodyne detection?

The main interest of using avalanche photodiode is to dramatically reduce the dark noise of the photodetector employed: our avalanche photodetectors present a dark noise around 50 counts per second, corresponding to a typical power of approx. 16 aW in the visible. With more traditional continuous detectors, one can hardly detect the shot noise level of a coherent state of less than 10 μ W. For lower optical powers, people usually employ homodyne readout techniques, which allow to go beyond this limitation (in that situation the LO power should be adjusted to bring the detection noise above the dark noise level, ie, more than 10 μ W). However the homodyne detection phase must be locked using interference fringes between the LO and the signal of interest, which in our case would differ in power by 10 orders of magnitude (100dB) (10 μ W for the LO and 1% (collection efficiency in reflexion) of the 100 fW injected for the signal), which is impossible to lock in practice (it is difficult to ensure an optical and electronic symmetry between the 2 arms of the homodyne detection better than 40 dB). The heterodyning technique also suffers from the same limitations on the symmetry, even if there is no fringe to lock.

Also, it is important to notice that it can be delicate to produce a LO shot noise limited in the kHz frequency range since there can be a large amount of technical noise coming from the laser source. On the contrary, operating with an extremely attenuated optical beam (our lasers provide 15 or 2 mW, and we attenuate them by 10 orders of magnitude typically), naturally reduces the technical noise down to insignificant levels, so that our sub pW fields are quasi perfectly at the shot noise level, as can be verified in the evolution of the background level in the power scans shown in the SI (Fig SI 7d)

Finally, the stability of the interferometer is the last important asset. Since the 2 interfering signals propagate along the same optical fiber and originate from 2 interfaces located in the dilution fridge, the fringe stability is exceptional. In practice, it is not even necessary to actively stabilize the fringes, except when one wants to scan the cryostat temperature above 10K.

3. Could the author label the equations in the Supplementary Information with sequential numbers? Readability would definitely benefit from that.

This has been done

4. Some additional recent references for

(i) low thermalization temperatures attained with probe rather than optical one: X. Zhou et al., Phys. Rev. Applied 12, 044066 (2019) DOI:10.1103/PhysRevApplied.12.044066

(ii) zeptoNewton force sensitivity for scanning probe technique: M. de Wit et al., Review of Scientific Instruments 90, 015112 (2019), DOI:10.1063/1.5066618

Those important references have been added. We also added:
Heritier et al, Nano Letters 18, 1814 (2018),
where good force sensitivities are also reported ($94 \text{ zN/Hz}^{0.5}$)

5. The optical measurement is performed on the light back-scattered by the nanowire into the input fiber.

We underline that one can also employ the transmission channel, where the signal to noise can be equivalent or even larger than in reflection (in our room temperature experiments, we obtain in general a SNB 10 times larger in transmission, but it suffers from a more delicate alignment: 2 objectives and the nanowire have to be properly aligned). By slightly displacing the transmission objective laterally (x direction), it can also provide a perpendicular readout channel, to track the nanowire vibrations along both transverse axes. However this channel will likely be unavailable in scanning probe measurements on the edge of a flat sample (due to a masking of the transmitted light by the spatially extended sample).

(i) How much is the efficiency with which the back-scattered light is re-coupled into the fiber?

The back-collection efficiency varies with the nanowire diameter and the light polarisation due to the Mie resonances, which structure the scattered field. As can be seen in the numerical simulations of Fig. 2a of the SI, it can reach up to approx. $0.5^2=0.25$ or 25 % in intensity. The measured experimental values are in good agreement with the simulations.

(ii) Regarding the back-scattered field, is it a Gaussian or a dipole-like field?

Due to the Mie resonances, the shape of the scattered field depends on the nanowire diameter: for an optical wavelength of 633 nm, it is dipolar only for diameters smaller than 100 nm typically, but higher order contributions rapidly come into play for larger diameters. Since the “quality factor” of the Mie resonances is pretty low (1-10), the scattered field rapidly becomes multipolar for larger diameters. The reflexion coefficients derived in the numerical simulations employed in Fig 2 of the SI are based on a calculation of the overlap between the complete scattered field with the counter-propagating illuminating mode. There we go beyond the paraxial and Gaussian approximation using the methodology of ref 3 of the SI (Iglesias et al) to describe the incoming and counter propagating reference fields.

6. The authors refer to their minimum level of optical flux used as the regime in which a single photon is collected per mechanical oscillation cycle. Is this a useful/particular regime in any sense?

This statement was simply meant to give a point of comparison to the reader and to underline the ultralow powers employed.

What is important is that the photon flux remains larger than the mechanical damping rate (10-100 mHz here). Indeed the Brownian motion can be roughly pictured as a succession of coherent oscillatory trajectories, with phase and amplitude resets every $1/Gm$ in time, so that 2 photons arriving at delays larger than $1/Gm$, will not present any correlations. In that situation, such $g(1)$ -like measurements will not operate (but one could employ $g(2)$ -like methods, such as the ones explored at high temperature in Mercier de Lépinay arxiv 2015).

7. At the end of page 4/beginning of page 5, the authors describe their fitting routine to estimate the noise temperature. I would like to have additional details and clarification about "testing a +/-10% variation of the damping rate around the best fit value". At the moment, the meaning of this sentence remains obscure to me.

When the signal to noise ratio is small, it can be delicate to properly estimate the quality factor of the mechanical resonance, and this may have some impact on the effective temperature given by the fit. To estimate this error, we follow the following procedure:

We realize a response measurement, which can be done with a smaller resolution bandwidth (3-20 mHz typically) and thus suffers less from the readout shot noise since it can be averaged over multiple frequency scans for example. The fit of the response gives us a pre-guess for the mechanical damping rate in the fit of the thermal noise spectra, which is in general very close to the fitted value (1-5% typically).

Then, to estimate the error in the fit of the effective temperature, we then realize 2 other fits, where we impose a +10%, -10% change on the mechanical damping rate (they are now fixed in the fits), and this gives us 2 bounds on the nanowire effective temperature. We then employ those 2 values as errors in the estimation of the effective temperature.

We have modified the sentence for clarity:

To evaluate the corresponding imprecision in the noise temperature, we fit the data by **imposing** a +/-10% variation of the damping rate around the best fit value, **giving two boundaries on the effective temperature**.

8. In the second column of page 5, the authors claim that "The temperature dependence of the mechanical frequency is found to be a very good indicator of the nanowire internal temperature...". Do the authors have additional data/reference to support this claim? Can they exclude any other mechanism to generate a similar change in the mechanical frequency?

The logarithmic increase of the mechanical frequency with temperature is a very robust phenomenology in the acoustic properties of amorphous materials at low temperatures (ref. 37 for example). It is termed "universal" in that field. It is a dispersive signature of the resonant (in opposition to the dissipative) interaction between the acoustic mode of interest and the TLSs that are quasi resonant with it. The pre-factor A in $df/f = A \ln(T/T_0)$ is proportional to the density of the TLS in the material and found in the $1e-4$ range for almost all amorphous resonators, see the new reference (PRL 100 195501, 2008) for example.

In our case, the nanowire are crystalline, and in principle it is only the surface of the nanowire that behaves amorphously, which can explain the lower density found (prefactor in the $1e-6$ range).

The word "found" in the sentence was misleading, this is not a discovery in itself, but simply a confirmation of the fact that what has been observed by many other groups was indeed applicable, with a different prefactor to our case. However this frequency dependence helped us identify and discriminate sources of artificial noise (which do not cause frequency shifts) from sources causing a "bulk" heating of the nanowire.

We have added a reference to Fefferman et al, PRL100,195501 (2008), where amorphous oscillators of similar frequencies were investigated on a large temperature range.

We have modified the sentence.:

“**We note that** the temperature dependence of the mechanical frequency **can be practically used** as a very good indicator...”

9. In the first column of page 6, around the end of the paragraph, the authors attribute the observed additional heating when probing the extremity of the nanowire to a wave guiding effect. Is this an hypothesis made by the authors? Can they back it up with a reference to existing literature? Could additional defects present at the nanowire tip explain the observed enhanced absorption?

We are currently running response measurements to investigate in greater details the specificities of the heat propagation in the nanowire using a fixed probe laser and a movable pump laser. A preliminary plot of the heating efficiency ((arb. units) measured at 300K with the probe laser when scanning the pump laser at the nanowire extremity is shown below: a significant increase appears in a region spanning over typically 1-2 μm from the nanowire tip. (see also the discussion with referee #1).

Heating efficiency, deduced from pump-probe response measurements, for different positions of the heating laser in the vicinity of the nanowire extremity (located on the right). The enhanced absorption is attributed to a wave guiding mechanism, which increases the amount of light which is channelled into the nanowire, and is thus more likely to induce optical absorption.

Similar behaviors were observed on all the nanowires we have investigated so far at room and dilution temperatures.

When moving the pump laser at different vertical locations, no similar local increase could be observed on the very homogeneous nanowire selected for the cryostat, so we believe it is not likely due to a localized surface contaminant.

Then, concerning the wave-guiding phenomenon: we can observe some very efficient wave guiding phenomena when we image on a CCD camera the light scattered out of the nanowire (by local scatterers such as dust or surface rugosity), while placing the laser at the nanowire extremity: the light can easily travel over tens of μm , but the mechanism is only present when the laser is positioned at the nanowire extremity and rapidly vanishes when the laser is placed

away from the nanowire tip. The phenomenon is polarisation dependent, as expected, and more contrasted for small nanowires, presenting a dipolar character.

CCD fluorescence images obtained in reflection (600-750 nm wavelength span) of a nanowire illuminated by a focussed 532 nm laser spot (500 nm spot size) positioned at the nanowire extremity (a) and approx 2 μ m above (b). The pump intensities and integration times are identical for both images. When the laser is positioned at the nanowire extremity, a fluorescence signal is observed along the nanowire, extending over tens of micrometers due to a wave-guiding mechanism. This is not the case, when the laser is positioned a few micrometers above the nanowire extremity. Note that the nanowire is tilted axially so that only its extremity is found sharp on the image. We employed a fluorescence imaging technique in order not to be blinded by the reflected pump beam.

As such, we are confident that the wave guiding mechanism plays an important role, it helps channelling the light inside the nanowire, and is thus likely to increase its chance of being absorbed.

We have added the above image to the SI, since it helps illustrating the wave-guiding mechanism.

10. At the beginning of page 7, the authors claim that "the absence of significant mechanical frequency shifts for increasing actuations strength suggests that the actuation mechanism does not heat up the nanowire.". However, before on page 5, second column, they say that the mechanical frequency shift is a good indicator for the internal temperature, and not for the mode temperature, which can be increased by parasitic force noise with negligible heat dissipation. How do the two statements coincide together?

Here we are afraid we do not really follow the referee question:

We first say that a mechanical frequency increase is an indicator of an increase of the bulk nanowire temperature (p5). Then we say page 7 that when we deal with an artificial noise increase (not bulk nanowire temperature) such as a fluctuating external force, the later increases the effective noise temperature of the nanowire, without changing neither the bulk nanowire temperature nor its frequency.

We then verify numerically that the power dissipated by the fluctuating force into the nanowire, even for very large artificial oscillation amplitudes, is extremely small (yW for a noise temperature of 1K), far smaller than the optical powers employed for example. This means that the fact that they do not lead to an appreciable temperature increase is not surprising in view of the very small dissipation rate of the nanowire.

Then, we state that the modulated force used to intentionally coherently drive the nanowire, does not generate neither an appreciable temperature increase, due to the ultralow mechanical dissipation rate. Maybe it was unclear that we were speaking of the coherent drive tone.

We have modified the sentence to:

“The absence of significant mechanical frequency shifts for increasing actuation strength suggests that the forces employed to coherently drive the nanowire do not increase its bulk temperature.”

11. Regarding the section about force sensing: experimentally, the authors resonantly drive one of the nanowire's eigenmode (with a relative large force of $\sim 2\text{aN}$), while they monitor the displacement of the other eigenmode, thus calculate the displacement spectral density to estimate the mode temperature. From this measurement, they estimate the force sensitivity of the driven eigenmode, that is they assume that both eigenmodes have the same mode temperature.

While the assumption is reasonable, the method seems a bit involved and I do not see the advantages compared to a more direct approach, in which the noise thermometry and the external force is applied on the same eigenmode. Could the author elaborate on this point? I think that a more convincing experiment for showing the force sensitivity would be to apply the smallest test force detectable in the displacement spectrum, with a signal-to-noise of around unity.

Concerning the mentioned measurement we indeed drive one eigenmode and monitor the displacement of the second mode, however we do not estimate the force sensitivity of the driven eigenmode, but the sensitivity of the non-resonantly driven eigenmode. The force sensitivity is deduced from a measurement of its thermal noise spectrum, to compute the thermal Langevin force noise (shown in Fig 5 and analyzed in the SI).

In general, we never assume that both modes present identical noise temperature, and we have indeed observed situations where the additional excess noise was not isotropic.

For what concerns the efficiency of the force drive, it of course depends on the projection of the driving force vector on the eigenvectors.

Concerning the single mode protocol mentioned by the referee:

the question that arises is for which resolution bandwidth such a measurement should be realized? As soon as the system operates in the linear regime and if the system stability is good enough, one can measure any arbitrarily small force drive by lowering the resolution bandwidth employed at the condition to acquire the signal for a sufficiently long duration (at least the inverse of the rbw).

The referee probably had in mind realizing a measurement with a rbw smaller than the mechanical linewidth, where the thermal noise would be well spectrally resolved, and where the force drive would appear as a tiny peak on top of it.

As a remark, in our case, we operate with resolution bandwidth often smaller than 10 mHz, to correctly resolve the thermal noise peak without flattening it by a convolution, so if one wants to detect a coherent drive on top of a well resolved thermal noise peak, one needs to operate with resolution bandwidths of a few mHz, which is the limit of our spectrum analyser.

We initially implemented such measurements, but at the end we were not much convinced by the conclusions that can be drawn from such measurement: one finally only checks that the noise outside of the drive tone frequency remains limited to the thermal noise level (which is

the case) but one does not check that the noise at the driving frequency is still limited by the Brownian motion. In that sense, the 2 mode measurement we employed suffers from the same limitation (see below the discussion on its importance for us).

Finally, a response measurement, such as the one shown in Fig. 5ab, based on demodulation technique, is probably more informative: the signal is measured within the demodulation bandwidth, and for the smallest driving tone, the measurement apparatus will only demodulate the nanowire thermal noise or the residual readout noise, which helps verifying that the demodulated signal amplitude follows a $\sqrt{\text{signal}^2 + \text{integrated noise power}}$ law, with an “offset” in agreement with the thermal noise spectra measured in absence of drive.

To illustrate this, we have complemented figure 5b by displaying the background of the fits, taken out of resonance, where one samples essentially the shot noise level, and by adding the demodulated noise background, indicated by the orange line (the SNR is around 1 in that measurement, so that the shot noise and mechanical signal contribute equally here), taken in absence of modulation (see the figure below).

Of course, it would have been better to have taken additional measurements at even lower drive amplitudes to illustrate the law discussed above, displayed in full line, but this was unfortunately not the purpose of the measurement at that time, where we aimed at verifying the linearity of the actuation/measurement chain, a prerequisite for employing the linear response theory.

Modified representation of fig 5b, where we have added the background of the response fits (squares). The thermal noise and shot noise contributions are shown in dashed line, using a resolution bandwidth of 50 mHz, while the total demodulated displacement amplitude signal is shown as a full line.

Then we conducted the 2 mode measurements because it was important for us to verify that our nanowires can also be operated as force gradient sensors, where the measurement proceeds through detection of eigenmode frequency shifts and eigenmode rotations (see our ref. 32). To allow realizing maps of external force fields, it is not realistic to employ measurement protocols based on thermal noise analysis in 2D due to the long acquisition time required. Instead, the only realistic method is to measure eigenmode rotations and eigenfrequency shifts using coherently driven trajectories, where a dual PLL serves to drive both transverse modes of the

nanowire. However one needs to verify that the ultralow noise properties of the force probes are preserved, even under large coherent oscillation amplitudes. This is the reason why we realized the 2 modes measurements and analyzed the frequency stability of the system under realistic drive conditions.

As such, we found that showing response measurements for increasing drive amplitude (Fig 5a,5b), which permits to assess the system linearity, combined with a noise analysis between the 2 perpendicular modes (Fig 5c, 5d) and a measurement of the frequency stability under external drive provided a more complete description of the force and force gradient measurement capacities of the nanowires in our experiment.

We believe we have clarified this point in the manuscript.

Reviewer #1 (Remarks to the Author):

From my view, the response to reviewers is quite satisfactory. I feel that the authors would pay more attention on evidence of material quality and structure in the revised manuscript/supplementary docs. I believe publications on Nature Communications would demonstrate the significant advances in the field and provide sufficient information for wide readership. The challenge in experimental physics is clearly addressed in this work, while understanding on material physics would need improvements in future studies. Overall, I would agree that the current presentation and explanation are quite adequate.

Reviewer #2 (Remarks to the Author):

The authors have addressed the points raised in my previous review. This manuscript is suitable for publication at this point.

Reviewer #3 (Remarks to the Author):

I thank the author for the thorough response to my previous questions. Before suggesting the paper for publications, I still have few questions that should be addressed. you can find my comments in the attached PDF as green-coloured text.

REVIEWER COMMENTS

Reviewer #1 (Remarks to the Author):

From my view, the response to reviewers is quite satisfactory. I feel that the authors would pay more attention on evidence of material quality and structure in the revised manuscript/supplementary docs. I believe publications on Nature Communications would demonstrate the significant advances in the field and provide sufficient information for wide readership. The challenge in experimental physics is clearly addressed in this work, while understanding on material physics would need improvements in future studies. Overall, I would agree that the current presentation and explanation are quite adequate.

Reviewer #2 (Remarks to the Author):

The authors have addressed the points raised in my previous review. This manuscript is suitable for publication at this point.

Reviewer #3 (Remarks to the Author):

I thank the author for the thorough response to my previous questions. Before suggesting the paper for publications, I still have few questions that should be addressed.
you can find my comments in the attached PDF as green-coloured text.

We thank the referees for their feedback on our work.

Please find below our response to the last comments raised by referee #3 (in bold black).

The changes in the manuscript and in the SI pdf files have been highlighted in yellow.

Best regards.

Remarks to the Author

1. A large fraction of the manuscript deals with the characterization of the mechanical properties in the low temperature regime, as well as a study on the heating mechanism due to optical absorption. Indeed, this is just an example of the physics one can study at these low temperature. In order to catch a larger audience and make the manuscript more visible, I would suggest the authors to rephrase the title and the abstract of the manuscript in order to englobe the thorough thermal characterization, which anyway represents a good part of the manuscript.

[...] We agree with the referee and we modified the title and the abstract to take into account the above mentioned aspects:

“Ultrasensitive nano-optomechanical force sensors operated at dilution temperatures”,

Cooling down nanomechanical force probes is a generic strategy to enhance their sensitivities through the concomitant reduction of their thermal noise and mechanical damping rates. However, heat conduction becomes less efficient at low temperatures, which renders difficult to ensure and verify their proper thermalization. We implement optomechanical readout techniques operating in the photon counting regime to probe the dynamics of suspended silicon carbide nanowires in a dilution refrigerator. Readout of their vibrations is realized with sub-picowatt optical powers, in a regime where less than one photon is collected per oscillation period. We demonstrate their thermalization down to 32 ± 2 mK, reaching record sensitivities for scanning probe force sensors, $40 \text{ zN/Hz}^{1/2}$, with a sensitivity to lateral force field gradients in the fN/m range. This opens the road toward explorations of the mechanical and thermal conduction properties of nanoresonators at

minimal excitation level, and to nanomechanical vectorial imaging of faint forces at dilution temperatures.

I would remove the sentence “..., in a regime where less than one photon is collected per oscillation period” (see below my reply to question 6).

Please refer below to our response to question 6.

5. The optical measurement is performed on the light back-scattered by the nanowire into the input fiber.

We underline that one can also employ the transmission channel, where the signal to noise can be equivalent or even larger than in reflection (in our room temperature experiments, we obtain in general a SNB 10 times larger in transmission, but it suffers from a more delicate alignment: 2 objectives and the nanowire have to be properly aligned). By slightly displacing the transmission objective laterally (x direction), it can also provide a perpendicular readout channel, to track the nanowire vibrations along both transverse axes. However this channel will likely be unavailable in scanning probe measurements on the edge of a flat sample (due to a masking of the transmitted light by the spatially extended sample).

I thank the authors for this remark. However, I should admit that I find surprising that the motion imprints a significant phase modulation on the transmitted field. For example, in the case of a movable semi-transparent mirror, only the phase of the reflected field gets modulated by the displacement of the mirror itself. For a dipolar scatterer moving in an optical field, the displacement information (for longitudinal motion) is mostly carried by the radiation scattered backward.

How does the authors' claim that the transmission channel offers equivalent or even higher SNR than the reflection channel match with my (probably naïve) picture explained above? Could the authors elaborate more?

The referee reasoning is correct for a nanowire and 2 objectives perfectly aligned along the optical axis: it is not possible to record any vibration signal at first order in this configuration in the transmission channel.

What we do experimentally is that we first carefully aligned the 2 objectives, and then we displace the transmission channel laterally, along x , by a fraction of the optical waist. In absence of nanowire, one measures a fix amount of light in the transmission channel that only depends on the lateral displacement applied (and decays as a Gaussian with x). When the nanowire enters the area between the 2 collection volumes, it is now capable of scattering part of the incident light into the transmission channel, so that we get a position dependent change of the transmission channel, as required to readout the nanowire dynamical vibrations.

Depending on the nanowire diameter and on the light polarisation, the amount of scattered light can be significant, in particular in the forward direction when higher order Mie resonances are at play, so that it is possible to obtain a decent lateral readout channel in transmission by doing so. When correctly aligned, the transmission channel can vary by $\sim 100\%$ on distances of a few 100 nm, which is a bit larger but comparable to what we have in the reflection channel ($\lambda/2$ periodicity), with the additional possibility to optimize the overall SNB by controlling the injected and collected polarisation states.

(as in reflection, the SNB depends on the slope of the position dependent signal and on the amount of light collected at the chosen measurement position (assuming it is shot noise limited, otherwise there can be additional technical noise))

The SNB strongly depends on the nanowire diameter: for small nanowires, the amount of scattered light is similar along all directions perpendicular to the induced optical dipole, but this is not the case when using higher order Mie resonances, which boost light scattering in the forwards direction. When using visible probe light, the nanowires employed are too large to behave as a simple dipole, and their scattering regime involves higher order Mie resonances which present a scattering diagram enhanced in the forward direction (see Bohren book). Due to its enhanced optical scattering, a nanowire can also almost totally suppress the transmission signal, even when collected on a large NA microscope objectives. When operating with quadrant photodiodes, in our room temperature experiment, we obtained a $>10\text{dB}$ larger SNB in transmission, on the transverse (x) measurement channel (see A. Glopped PhD thesis, 2014).

In practice, it is more delicate to align 2 objectives and the nanowire, and in a typical force sensing experiment, we do not necessarily have access to the transmission channel: depending on the geometry of the sample investigated, it can be hidden (or become dependent on the sample position, which is annoying since it requires a permanent re-estimation of the measurement slope). As such our measurements were mainly done through the reflection channel.

More generally, to fully investigate 2D force fields it is necessary to combine reflection and transmission channel to operate with a complete 2D readout (Mercier de Lépinay 2018).

6. The authors refer to their minimum level of optical flux used as the regime in which a single photon is collected per mechanical oscillation cycle. Is this a useful/particular regime in any sense?

This statement was simply meant to give a point of comparison to the reader and to underline the ultralow powers employed.

What is important is that the photon flux remains larger than the mechanical damping rate (10-100 mHz here). Indeed the Brownian motion can be roughly pictured as a succession of coherent oscillatory trajectories, with phase and amplitude resets every $1/Gm$ in time, so that 2 photons arriving at delays larger than $1/Gm$, will not present any correlations. In that situation, such $g(1)$ -like measurements will not operate (but one could employ $g(2)$ -like methods, such as the ones explored at high temperature in Mercier de Lépinay arxiv 2015).

I agree with the authors about the importance of having a photon flux faster than the mechanical damping rate for the kind of measurements performed.

However, I do not like the comparison between the optical flux and the mechanical resonance frequency. The reason is that such a comparison suggests that there is something special about having photon fluxes lower (or higher) than the mechanical resonance frequency. For example, in the context of measurement theory, comparisons with the mechanical resonance frequency are usually done in order to highlight different measurement regimes (e.g., weak vs strong measurements).

On the other hand, I understand the authors' need to stress the extremely low employed power.

I would be happy if the authors remove the sentence "..., in a regime where less than one photon is collected per oscillation period" from the abstract, and if they make clear in the introduction that this serves just as a comparison (substituting the sentence "..., while our measurements are realized with less than one photon detected per mechanical period" with something along the line "To give a sense of the ultralow power used, such a photon flux corresponds to...").

Since we operate in the photon counting regime (which is an experimental novelty of the work), we find that it is important to underline the weakness of the flux employed,

and if one should not compare it to the weak/strong measurement, the fact is that we get information on the oscillator position less than once per oscillation period.

We find that this is an important remark to underline in the manuscript.

In order to not unbalance the abstract, we propose to remove the term "regime" which can give the impression that we underline an important theoretical aspect, to the more neutral term: "situation": "... in a situation where less than one photon is collected..."

Then, in the main text, we have modified the sentence:

Instead, we have developed a methodology based on avalanche single photon counters operated in the Geiger ('click') mode featuring ultra-low dark count rates in the attowatt range, while our measurements are realized with collected photon fluxes of a few 10^3 counts.s⁻¹ (less than one photon detected per mechanical period).

7. At the end of page 4/beginning of page 5, the authors describe their fitting routine to estimate the noise temperature. I would like to have additional details and clarification about "testing a +/-10% variation of the damping rate around the best fit value". At the moment, the meaning of this sentence remains obscure to me.

When the signal to noise ratio is small, it can be delicate to properly estimate the quality factor of the mechanical resonance, and this may have some impact on the effective temperature given by the fit. To estimate this error, we follow the following procedure:

We realize a response measurement, which can be done with a smaller resolution bandwidth (3-20 mHz typically) and thus suffers less from the readout shot noise since it can be averaged over multiple frequency scans for example. The fit of the response gives us a preguess for the mechanical damping rate in the fit of the thermal noise spectra, which is in general very close to the fitted value (1-5% typically).

Then, to estimate the error in the fit of the effective temperature, we then realize 2 other fits, where we impose a +10%, -10% change on the mechanical damping rate (they are now fixed in the fits), and this gives us 2 bounds on the nanowire effective temperature. We then employ those 2 values as errors in the estimation of the effective temperature.

We have modified the sentence for clarity:

To evaluate the corresponding imprecision in the noise temperature, we fit the data by **imposing** a +/-10% variation of the damping rate around the best fit value, **giving two boundaries on the effective temperature.**

I understand now the analysis procedure used by the authors. However, it triggers a new question: is the amount of variation chosen (10%) arbitrary? Does it reflect the statistical error in the measured spectra?

In a model comprising a Lorentzian function and a flat background (as in Eq. (1)), the effective temperature parameter represents the area under the peak (that is, the area after the background has been subtracted). If proper weights are given to each fitted spectral components, I would assume that the confidence interval from the fitting procedure reflects the uncertainty of the fit in estimating the free parameters.

In general, the smaller the SNR the larger this confidence interval for the area. Have the author investigated the fit confidence interval? Is it consistent with the 10% variation the impose on the (now fixed) damping rate parameter?

The initial goal of the discussion was to give the reader an idea of the convergence of the fit function, which can be critical when the SNB is too small. In the attached image, we present the convergence plot of the fit function for different damping rates, which was previously used in our discussion.

Convergence plot of the normalized quadratic error of the fits (for the coldest measurement shown in Fig. 3), as a function of the effective temperature for different damping rates (-10, -5, 0, +5, +10 % change with respect to the best value (53.8 mHz). The smallest error values for the different traces are obtained for 30.4, 31.2, 32.06, 32.8, 33.5 mK respectively. The horizontal lines represent a 0%, +5% and +10 % change with respect to the minimum normalized quadratic error (~0.15%). The shot noise level, which can be averaged on many frequency points far from resonance, is fixed for all fits.

The statistical error we obtained while repeating thermal noise measurements were of +/- 5% for the damping rates and a similar value for the effective temperature. From the above analysis, we see that those values are in agreement with the convergence fit on a single measurement. In the manuscript, we used a larger value (+-10% error on Gm) in order not to underestimate the confidence band for our measurement, but we agree that this was confusing.

As such we have modified the presentation of the error to the following progression, where for clarity we only discuss the statistical errors (after having added to the SI for convergence plot of the fit function):

At a given position within the interference pattern, if we repeat thermal noise spectrum measurements, the statistical error we obtain on the effective temperature is around $\pm 5\%$. As such this would give a statistical confidence interval of 30.5 - 33.7 mK around the mean value (32.06 mK).

Then, as stated in the manuscript, the error we have in the measurement of the readout slope, which is recorded in our algorithm, is responsible for a $\pm 2.5\%$ variation of the calibrated displacement noise power, which has to be added to the previous statistical error (quadratically since those are 2 independent mechanisms).

The total statistical error is then $\sqrt{5\%^2 + 2.5\%^2} \sim 6\%$. As such we now converge to an estimation of 32.1 ± 1.8 mK effective temperature. We now state this value in the main text, but round it to 32 ± 2 mK in the abstract.

The main text has been modified to:

"Due to the modest signal to background ratio obtained, it can be delicate to properly estimate the nanowire quality factor (but it can be better evaluated using driven measurements, see below). To evaluate the corresponding imprecision in the noise temperature evaluation, we fit the data by imposing a $\pm 5\%$ variation of the damping rate around the best fit value, giving a variability of $\pm 5\%$ on the fitted temperature (see SI). Those values are in agreement with the statistical error we obtained on iterative measurements, while the overall absolute imprecision is limited by the one obtained on the measurement slope to a $\pm 2.5\%$ level. As a result, we estimate the effective temperature of the nanowire vibration noise at a level of 32.1 ± 1.8 mK, for a sample temperature of 27 ± 3 mK, measured with a RuO₂ thermometer (and a 22 mK mixing chamber temperature)."

8. In the second column of page 5, the authors claim that "The temperature dependence of the mechanical frequency is found to be a very good indicator of the nanowire internal temperature...". Do the authors have additional data/reference to support this claim? Can they exclude any other mechanism to generate a similar change in the mechanical frequency?

The logarithmic increase of the mechanical frequency with temperature is a very robust phenomenology in the acoustic properties of amorphous materials at low temperatures (ref. 37 for example). It is termed "universal" in that field. It is a dispersive signature of the resonant (in opposition to the dissipative) interaction between the acoustic mode of interest and the TLSs that are quasi resonant with it. The pre-factor A in $df/f = A \ln(T/T_0)$ is proportional to the density of the TLS in the material and found in the $1e-4$ range for almost all amorphous resonators, see the new reference (PRL 100 195501, 2008) for example. In our case, the nanowire are crystalline, and in principle it is only the surface of the nanowire that behaves amorphously, which can explain the lower density found (prefactor in the $1e-6$ range).

The word “found” in the sentence was misleading, this is not a discovery in itself, but simply a confirmation of the fact that what has been observed by many other groups was indeed applicable, with a different prefactor to our case. However this frequency dependence helped us identify and discriminate sources of artificial noise (which do not cause frequency shifts) from sources causing a “bulk” heating of the nanowire.

We have added a reference to Fefferman et al, PRL100,195501 (2008), where amorphous oscillators of similar frequencies were investigated on a large temperature range.

We have modified the sentence.:

“We note that the temperature dependence of the mechanical frequency can be practically used as a very good indicator...”

As the authors pointed out, the behaviour is universal for *amorphous* materials, whereas they have a *crystalline* material surrounded by an *amorphous* layer. Are the authors assuming that the crystallin bulk material behaves similarly to an amorphous one, although with a lower density of TLSs?

We do not make or use such an assumption.

The temperature dependence of the frequency and mechanical dissipation in Silicon Carbide have not yet been measured in the low temperature range investigated (to our best knowledge). However if one compares to other crystalline materials (quartz or silicon in particular), the relative frequency shifts or damping changes are significantly smaller in this temperature range compared to what has been observed in the amorphous phase of identical materials (see Enss and Hunklinger book for example). Furthermore the mechanical properties of crystalline materials do not feature the peculiar temperature dependence observed and instead, monotonously converge towards zero at low temperatures.

If one assumes that the dissipation in the crystalline part of the nanowire is negligible, then we are essentially limited by the amorphous crust contribution. As such the effective TLS density (rescaled to the entire nanowire volume) would be reduced since only part of the nanowire is amorphous and contributes to the mechanical frequency shifts and dissipation rate. We do not go beyond this simple observation. Completely understanding the dissipation in a core –shell system at such low temperatures would require a far more involved dedicated investigation.

We thank the referee again for the comments on our work.

Reviewer #3 (Remarks to the Author):

The authors have given satisfactory and detailed answer to all my questions.
At this point, I can advise publication of the present manuscript in Nature Communications.
All the best.